# When Glass Disappears at Night: A Novel NIR-RGB Multi-modal Solution

**Tao Yan**[1,*]**, Yiwei Lu**[1]**, Ke Xu**[3]**, Hao Chen**[1]**, Hui Li**[1]**, Xiaojun Chang**[2]**, Xiaojun Wu**[1]**, Rynson W.H. Lau**[3]

[1] *School of Artificial Intelligence and Computer Science, Jiangnan University*

[2] *School of Information Science and Technology, University of Science and Technology of China*

[3] *Department of Computer Science, City University of Hong Kong*

*Correspondence to: yantao.ustc@gmail.com*

**Reviewed on OpenReview:** *https://openreview.net/forum?id=hdh3vHsakv*

## Abstract

Glass surface detection (GSD) has recently been attracting research interests. However, existing GSD methods focus on modeling glass surface properties for daytime scenes only, and can easily fail in nighttime scenes due to significant lighting discrepancies. We observe that, due to the spectral differences between Near-Infrared (NIR) light sources and common LED lights, NIR and RGB cameras capture complementary visual patterns (*e.g.*, light reflections, shadows, and edges) of glass surfaces, and cross-comparing their lighting and reflectance properties can provide reliable cues for nighttime GSD. Inspired by this observation, we propose a novel approach for nighttime GSD based on the multi-modal NIR and RGB image pairs. We first construct a nighttime GSD dataset, which contains $6,192$ RGB-NIR image pairs captured in diverse real-world nighttime scenes, with corresponding carefully-annotated glass surface masks. We then propose a novel network for the nighttime GSD task with two novel modules: (1) an RGB-NIR Guidance Enhancement (RNGE) module for extracting and enriching the NIR reflectance features with the guidance of RGB reflectance features, and (2) an RGB-NIR Fusion and Localization (RNFL) module for fusing RGB and NIR reflectance features into glass features conditioned on the multi-modal illumination discrepancy-aware features. Extensive experiments demonstrate that our method outperforms state-of-the-art methods in nighttime scenes while generalizing well to daytime scenes. Our dataset and code are available at `https://github.com/YT3DVision/NGSDNet`.

## 1 Introduction

Glass surfaces, such as glass doors, walls, and windows, are ubiquitous in our daily lives. Their lack of intrinsic visual texture patterns can easily conceal glass surfaces within the background scene, causing significant detection difficulties. Failing to detect glass surfaces may cause the downstream vision applications, including 3D scene reconstruction and robotic navigation, to fail as well. Hence, glass surface detection (GSD) is a challenging, but fundamental task.

A few deep learning-based methods are proposed to detect glass surfaces based on RGB features (Mei et al., 2020; He et al., 2021; Lin et al., 2021; Fan et al., 2023; Liu et al., 2024; Lin et al., 2022; Yan et al., 2025; Qi et al., 2024; Cheng et al., 2026), or RGB-X multi-modal features (Lin et al., 2025; Huo et al., 2023; Yan et al., 2024). These methods typically focus on modeling different priors for detecting glass surfaces, including contrasted RGB (Mei et al., 2020) or RGB-Thermal (Huo et al., 2023) features, boundaries (He et al., 2021; Fan et al., 2023), reflections (Lin et al., 2021; Liu et al., 2024; Yan et al., 2024), perceived noisy depth (Lin et al., 2025), semantic correlations (Lin et al., 2022; Cheng et al., 2026), and ghosting effects (Yan et al., 2025). However, these GSD priors are specifically developed for daytime scenes and can be drowned in the low-light

*Corresponding authors

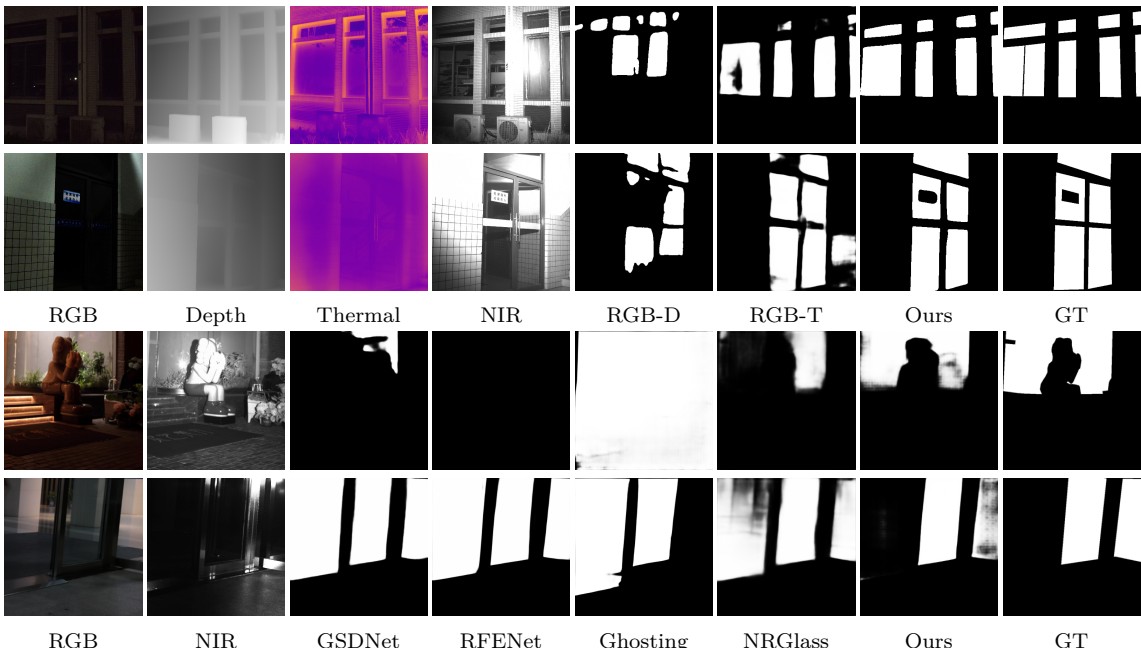

Figure 1: Upper two rows: Depth (Lin et al., 2025) and thermal (Huo et al., 2023) cues provide very limited contextual information for localizing glass surfaces in nighttime scenes, compared to NIR imaging. Bottom two rows: Intrinsic cues such as reflections (in either RGB (Lin et al., 2021) or RGB-NIR (Yan et al., 2024) domain), and boundary (Fan et al., 2023) and ghosting effects (Yan et al., 2025) (in the RGB domain), can be easily buried in nighttime scenes. We propose to cross-compare the lighting and reflectance information between RGB and NIR modalities for accurate glass surface detection in nighttime scenes.

or complex artificial lighting of nighttime scenes. For example, as shown in the top two rows of Fig. 1, compared to NIR imaging (wavelength typically ranging from $0.76\mu$m to $3\mu$m), depth or thermal imaging (wavelength typically ranging from $8.0\mu$m to $14\mu$m) provides limited contextual information in nighttime scenes for existing RGB-depth/thermal-based GSD methods (Lin et al., 2025; Huo et al., 2023) to detect glass regions. Meanwhile, modeling intrinsic cues such as boundary (Fan et al., 2023), reflections (Lin et al., 2021), and ghosting effects (Yan et al., 2025) for glass surface localization in the RGB domain (Lin et al., 2021; Fan et al., 2023; Yan et al., 2025), or comparing reflections between the RGB and NIR modalities (Yan et al., 2024) is unreliable in nighttime scenes, as demonstrated in the bottom two rows of Fig. 1.

We observe that NIR and RGB cameras can capture complementary visual cues (*e.g.*, reflection/transmission discrepancies and illumination discrepancies) for nighttime GSD. By projecting their own lights, active NIR cameras ensure consistent visibility in low-light/uneven lighting conditions, producing geometry/reflectance patterns on glass surfaces that complement those (*e.g.*, colors, textures, and semantics) in the RGB modality.

However, RGB-NIR image fusion and detection have been sparsely studied. Numerous RGB-Infrared (Thermal) image fusion methods strive for quality improvement but leave alone the follow-up detection, and the fusion emphasizes more on "seeking commons" but neglects the differences of these two modalities (Liu et al., 2022a). Therefore, in this paper, we propose a novel RGB-NIR-based approach, which considers the complementary patterns of glass surfaces between NIR and RGB images, for nighttime GSD.

As there are no available datasets for this task, we first construct a large-scale RGB-NIR glass surface detection dataset, with a hybrid imaging system consisting of a DSLR camera and an NIR camera accompanied by an active NIR light source.[1] Our dataset contains $6,192$ RGB-NIR image pairs captured from diverse real-world nighttime scenes, with the corresponding manually annotated glass surface masks. We then propose a novel neural network to model the complementary patterns on glass surfaces between NIR and RGB images for glass surface detection.

---

[1]Nighttime surveillance systems always use active NIR cameras accompanied by active NIR light sources.

To extract the complementary patterns between RGB and NIR images as cues, our method first performs a learning-based image decomposition to decompose the input RGB and NIR images into two pairs of reflectance and illumination components, and then uses two separate encoders to extract the semantics-aware and material-aware contextual features from the reflectance components of the two modalities. We further introduce two novel modules. First, we propose a novel RGB-NIR Guidance Enhancement (RNGE) module, which exploits the rich semantic features in the RGB reflectance component to refine and enrich the feature representation of its NIR counterpart. In other words, the RNGE module aims to explore the semantics from the RGB image to assist glass feature extraction, especially boundary prediction, from the NIR image. Meanwhile, we model the illumination discrepancies between RGB and NIR images as gating matrices based on their derived Illumination components, and propose a novel RGB-NIR Fusion and Localization (RNFL) module for decoding the multi-modal reflectance features into glass features conditioned on the derived gating matrices. As shown in Fig. 1, our method can produce more accurate detection results under challenging lighting conditions of night-time scenes. Our method can be deployed on popular surveillance cameras that switch between RGB and active NIR modes. The main contributions of this work can be summarized as follows:

- We propose the first approach for nighttime glass surfaces detection, by modeling the complementary patterns of glass surface regions between RGB and NIR image pairs.

- Our network comprises two novel modules: an RNGE module for enriching NIR reflectance features with RGB reflectance features, and an RNFL module for GSD based on multi-modal reflectance feature aggregation guided by the multi-modal illumination differences.

- We construct the first large-scale nighttime GSD dataset, which contains $6K$ RGB-NIR glass image pairs (with corresponding masks) captured from diverse nighttime scenes.

- Extensive evaluations show that our proposed method outperforms SOTA methods in nighttime scenes while maintaining competitive performance in daytime conditions.

## 2 Related Work

**Glass Surface Detection** (GSD) has recently gained significant research attention with several deep learning-based methods proposed. A line of GSD methods are based on input RGB frames, which model the contrasted contextual features (Mei et al., 2020) and incorporate reflection priors (Lin et al., 2021; Liu et al., 2024), boundary detection (He et al., 2021; Lin et al., 2021; Fan et al., 2023), semantic correlations (Lin et al., 2022; Cheng et al., 2026), blurry effects (Qi et al., 2024), and ghosting cues (Yan et al., 2025). Another line of methods explore RGB-X multi-modal imaging for GSD. Kalra et al. (2020) leverages the polarization information to segment transparent objects (*e.g.*, wine glass and glass balls), which may not generalize well to glass surfaces with irregular shapes. Huo et al. (2023) models the contrasted glass features between RGB and Thermal modalities. Yan et al. (2024) model the reflection differences between RGB and NIR images for daytime GSD. They use an NIR filter attached to the DSLR camera lens and rely on the ambient light to capture NIR images. Lin et al. (2025) propose to model the noise differences between RGB and depth images. Most recently, Zhang et al. (2025) propose the MonoGlass3D method, performing 3D glass segmentation and 3D plane regression simultaneously.

All these existing methods, however, are designed for daytime scenes, and their proposed cues may not be effective under low-light or complex artificial lighting conditions of nighttime scenes. In this work, we propose to detect glass surfaces at nighttime scenes by utilizing dual RGB-active NIR cameras. In contrast to passive NIR filters (Yan et al., 2024), where both RGB and NIR images may not be sufficiently illuminated in nighttime scenes, our imaging setup creates an induced photometric discrepancy (from active NIR illumination and ambient lighting), and our network learns to analyze this discrepancy for the detection.

**Mirror Detection** (MD) aims to detect mirror regions. Existing methods focus on learning the correlations between the reflected and real surrounding contents, by modeling the contextual contrasted RGB (Yang et al., 2019; Xu et al., 2024) or RGB-Depth (Mei et al., 2021b) features, pearance correspondences (Lin et al., 2020;

2023; Huang et al., 2023), visual chirality (Tan et al., 2023), spatial/frequency-based specular textures (Xie et al., 2024), and inconsistent motions (Warren et al., 2024; 2026).

Although these methods have achieved impressive progress for mirror detection, glass surfaces have a very different property from mirrors. While mirrors only contain reflection, glass surfaces contain both reflection as well as transmission, making glass surfaces more challenging to detect. In this paper, we explore the use of RGB-NIR images for glass surface detection.

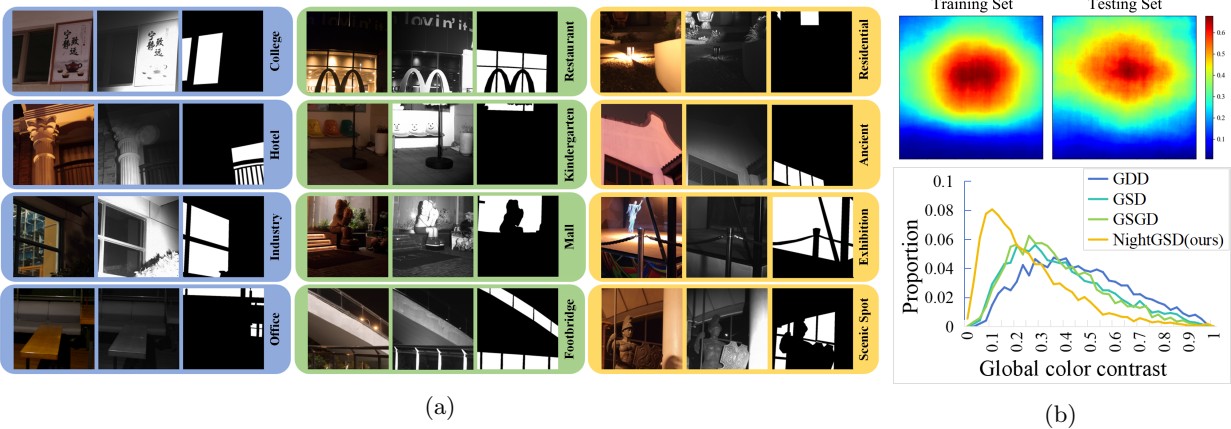

(a)      (b)

Figure 2: Examples and some statistics of our dataset: (a) Example RGB/NIR/Glass mask triplets from our dataset; (b) Glass location distributions (top) and Color contrast distribution (bottom).

## 3  Proposed Dataset

We construct the first large-scale nighttime glass surface detection dataset for training and evaluation.

**Hybrid Imaging System.** We design a hybrid imaging system consisting of a DSLR camera (Canon 70D) and an NIR camera (HIKVISION MVCH250-90GN) with an active NIR light source (40W). Both cameras are synchronized to capture RGB and NIR image pairs from the target scene simultaneously. Although the reflectivity of most glass surfaces decreases with increasing wavelength of the incident ray within the range of $780 \sim 1100$nm (Huo et al., 2023; Planinsic, 2011), excessively long wavelengths will diminish night vision effectiveness (Ariff et al., 2015). Therefore, in our imaging system, the wavelength of the NIR light (emitted by the active infrared light source) is aligned with the camera's fixed spectral sensitivity, operating in the $850 \sim 940$nm band (Ariff et al., 2015). This selection balances NIR reflection suppression and night vision capability, enabling clearer NIR images with reduced glass surface reflection. We use the binocular image alignment method (Shen et al., 2020) to align each RGB-NIR image pair.

**Dataset Statistics.** Our dataset consists of $6,192$ triplets of RGB and NIR images and the corresponding manually labeled glass surface masks. Our dataset covers 12 types of daily life scenes, such as campus, hotel, and shopping mall, as shown in Fig. 2a. We collected these images both indoors and outdoors. Notably, most glass surfaces in the dataset are smooth and planar, which reflects common real-world configurations but may introduce bias against curved structures. For training and evaluation, we randomly split our dataset into $5,000$ and $1,192$ triplets, respectively. Fig. 2b shows the statistics of glass location distribution and the color contrast distribution over the image. Glass regions predominantly appear in the upper image region due to their typical placement at or above eye level, potentially limiting generalization to atypical configurations. The color contrast distribution (Lin et al., 2021) reflects the similarity between glass and non-glass regions across the dataset. Specifically, we compute the $\chi^2$ distance between glass regions and non-glass regions in each image to measure their contrast, forming the color contrast distribution for the entire dataset. In our nighttime GSD dataset, most samples fall within the $[0, 0.4]$ range, indicating a high similarity between glass and non-glass regions, which makes glass surface detection more challenging.

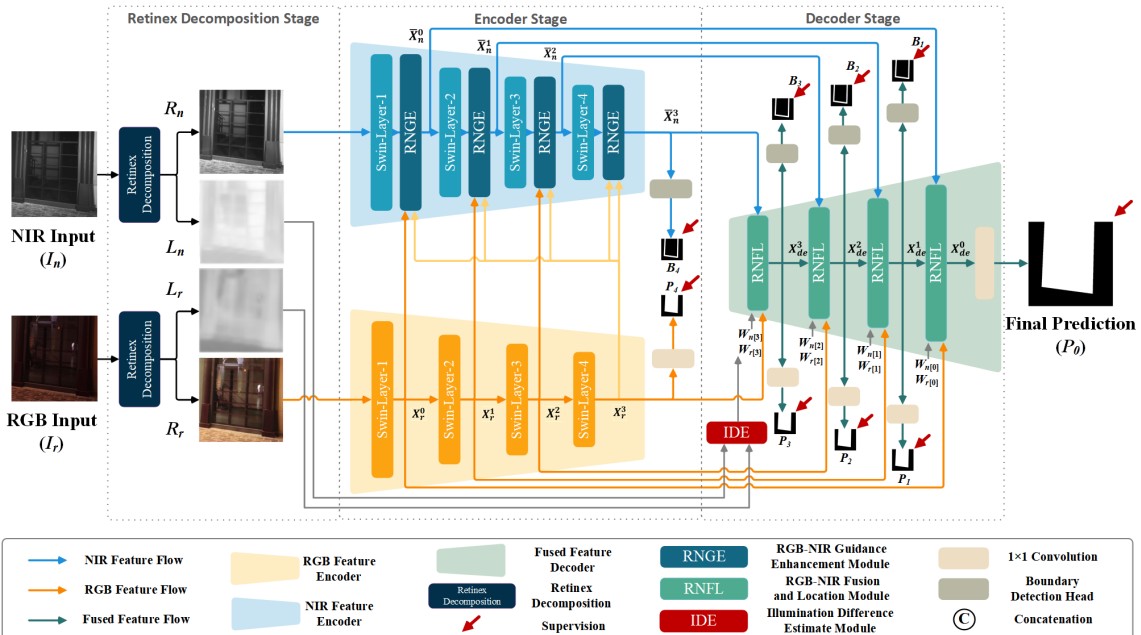

Figure 3: Method Overview. The NIR ($I_n$) and RGB ($I_r$) images captured at nighttime are first decomposed into corresponding reflectance ($R_n$ and $R_r$) and illumination ($L_n$ and $L_r$) layers. We use two encoders to process the two reflectance layers with a novel RNGE module to guide material-aware features extraction from the NIR reflectance $R_n$. We cross-compare the illumination layers ($L_n$ and $L_r$) via the IDE module to control the feature flow in the decoder, where we propose the RNFL module to selectively fuse the dual-modality features for glass surface detection.

## 4 Proposed Method

Our core idea is to cross-compare and fuse the lighting and material information captured in the RGB and NIR modalities for nighttime glass surface detection. Fig. 3 shows an overview of our method, which includes the Retinex Decomposition, the Encoder, and the Decoder stages.

The Retinex Decomposition stage aims to extract lighting and reflectance information from the input multi-modal images, since the imaging of both RGB and NIR images is dependent on an external light source. We fine-tune the SOTA Content-Transfer Decomposition Network (CTDN) (Jiang et al., 2024), which performs Retinex decomposition in the latent space to obtain content-rich reflectance components and content-free illumination components. The CTDN decomposes the input RGB image $I_r$ into Reflectance component $R_r$ and Illumination component $L_r$, and the input NIR image $I_n$ into Reflectance component $R_n$ and Illumination component $L_n$.

The Encoder stage aims to extract the semantics-aware features from the RGB reflectance component, which guides and enhances the material-aware feature extraction from the NIR reflectance component. We use the Swin Transformer V2 (Liu et al., 2022b) to extract multi-scale features (denoted as $X_r^i$ and $X_n^i$, where $i \in \{0, 1, 2, 3\}$) from the two Reflectance components ($R_r$ and $R_n$), respectively. At each scale, we propose the RNGE module (Sec. 4.1) to leverage the high-level semantic features $X_r^3$ and the low-level features $X_r^i$ to guide and enhance the extraction of features $X_n^i$ from $R_n$. Meanwhile, since the NIR reflectance component $R_n$ exhibits clearer and more complete boundaries of glass surfaces than those in $R_r$, we perform the boundary detection on the deepest features $\bar{X}_n^3$. Notably, for handling extreme low-light conditions, our method is designed to rely on the active NIR modality.

The Decoder stage aims to transform the multi-modal features into glass surface features for detection. We first model the illumination differences between the NIR and RGB images via the Illumination Difference Estimation (IDE) module (Sec. 4.2), which takes the illumination components of two modalities ($L_n$ and $L_r$)

as input and predicts two weight matrices $W_r$ and $W_n$ for controlling the feature flows in the decoder. We then propose the RGB-NIR Fusion and Localization (RNFL) module (Sec. 4.3), which works at multi-scales, integrating the extracted multi-modal features $R_r$ and $R_n$ from the Encoder stage with guidance from the IDE module for predicting the glass surface masks.

## 4.1 RGB-NIR Guidance Enhancement (RNGE) Module

The proposed RNGE module aims to extract and enhance multi-scale NIR features from $R_n$ conditioned on corresponding RGB features. As shown in Fig. 4, we first concatenate $X_n^i$ and $X_r^i$, and then use a convolution layer to produce the fused features $X_f^i$. Meanwhile, we compute the difference between $X_n^i$ and $X_r^i$ as $X_d^i$ through element-wise subtraction. Since $X_d^i$ highlights modality-specific discrepancies, which may provide informative cues for glass surface localization, we apply a convolution layer and a sigmoid($\cdot$) activation to normalize $X_d^i$, yielding the activation map $M_d$.

We then multiply $X_f^i$ by $M_d$ to obtain the enriched glass features $\bar{X}_f^i$. In addition, we apply supervision to $X_r^3$ (as shown in Fig. 3) to capture abundant semantic features of glass surfaces and use them as guidance to the multi-modal feature fusion. Specifically, $X_r^3$ is first upsampled to the size of $X_n^i$ and normalized by a sigmoid($\cdot$) function to produce the activation map $M_c$, which is then applied to $\bar{X}_f^i$ for further enrichment. Finally, we use a self-attention (SA) block (Vaswani, 2017) to enhance the glass features and transform them back to the initial $X_n^i$ through a residual connection to produce the output features $\bar{X}_n^i$. The whole process can be formulated as:

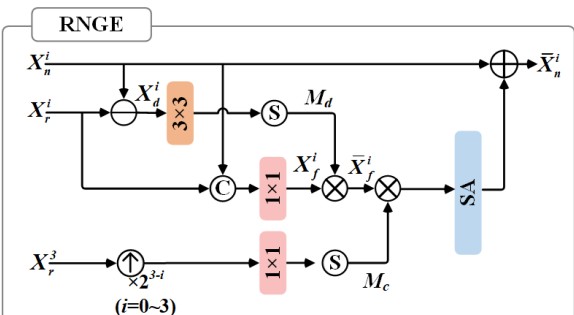

Figure 4: The proposed RNGE module.

$$
\begin{aligned}
X_f^i &= \text{Conv}(\text{Concat}(X_n^i, X_r^i)), \\
\bar{X}_f^i &= X_f^i \otimes \text{sigmoid}(\text{Conv}(X_n^i - X_r^i)), \\
\bar{X}_n^i &= \text{SA}(\bar{X}_f^i \otimes \text{sigmoid}(\text{Conv}(\text{Up}(X_r^3)))) + X_n^i.
\end{aligned}
\tag{1}
$$

Fig. 5 shows the input features $(X_n^0)$, the difference features $(X_d^0)$, the fused features $(X_f^0)$, the activation map $M_c$ for enrichment, the contrasted activation map $M_d$ between the two modalities, and the enhanced features $(\bar{X}_n^0)$ by the RNGE module.

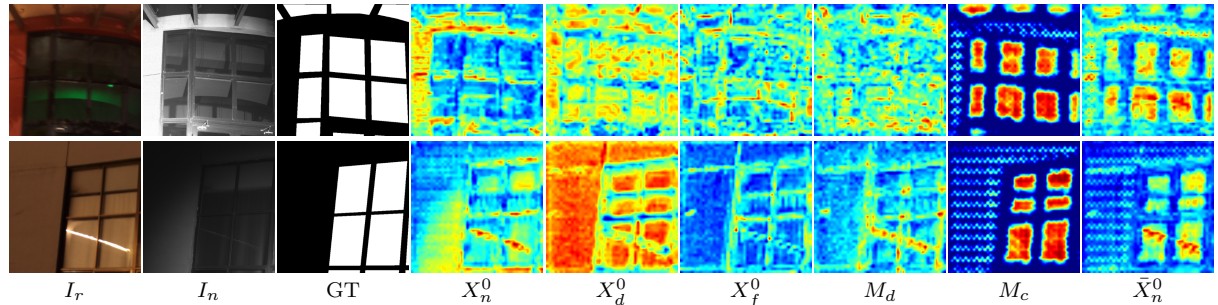

| $I_r$ | $I_n$ | GT | $X_n^0$ | $X_d^0$ | $X_f^0$ | $M_d$ | $M_c$ | $\bar{X}_n^0$ |

Figure 5: Intermediate feature visualization of the RNGE module.

## 4.2 Illumination Difference Estimation (IDE) Module

The IDE module, as shown in Fig. 6, aims to guide the fusion of the features of the RGB and NIR modalities, by modeling the difference in illumination components between the two modalities through estimating two gating matrices $W_r$ and $W_n$. Specifically, to accurately measure the difference between $L_n$ and $L_r$, both of them are first fed into the $1 \times 1$ Average Pooling (AP) block followed by two $5 \times 5$ convolution layers to produce refined features $L_n'$ and $L_r'$. The difference $L_d$ between them is then computed using element-wise subtraction to

capture illumination variations across different areas of the two modalities. We further leverage $L_d$ to enhance the different areas in the original RGB-NIR image pair by concatenating $L_d$ with $L_n'$ and $L_r'$ separately, and subsequently adjusted to 4 channels via two $3 \times 3$ convolution layers, matching the four scales of the Decoder. For each branch, we use a spatial-attention block (Woo et al., 2018) to further enhance the corresponding features and a sigmoid($\cdot$) function to normalize these feature values to the range of $[0, 1]$, producing two weight matrices $W_n$

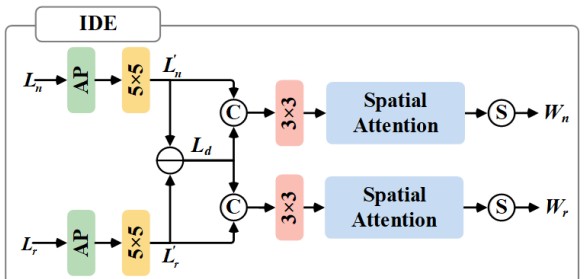

Figure 6: The proposed IDE module.

and $W_r \in R^{W \times H \times 4}$, where $[W, H]$ is the spatial resolution of the input images. The whole process can be described as:

$$
\begin{aligned}
L_d &= \text{Conv}(\text{AP}(L_n)) - \text{Conv}(\text{AP}(L_r)), \\
W_n &= \text{sigmoid}(\text{SPA}(\text{Conv}(\text{Concat}(L_n', L_d)))), \\
W_r &= \text{sigmoid}(\text{SPA}(\text{Conv}(\text{Concat}(L_r', L_d)))),
\end{aligned}
\tag{2}
$$

where SPA($\cdot$) denotes the spatial attention block (Woo et al., 2018).

As the weight matrices $W_r$ and $W_n$ visualized in Fig. 7, the IDE module learns to prioritize deeper RGB features and shallower NIR features. This is because RGB contains richer semantic information, while NIR provides more distinct edge and structural details.

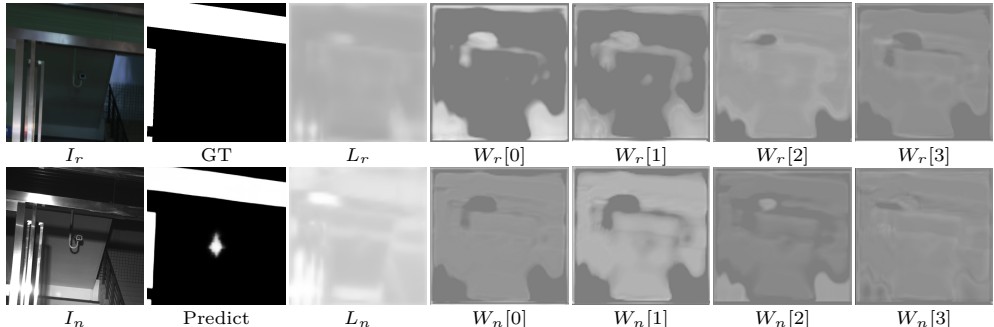

Figure 7: Visualization of weight matrices $W_r$ and $W_n$. Larger intensities indicate higher weights.

### 4.3 RGB-NIR Fusion and Localization (RNFL) Module

The proposed RNFL module, as shown in Fig. 8, aims to effectively aggregate multi-modal features to localize glass surfaces, which contains two groups of cross-attention (CA) mechanisms, *i.e.*, the left and right parts. The left two cross-attention mechanisms aim to extract the shared glass features from $X_r^i$ and $\bar{X}_n^i$.

Specifically, $\bar{X}_n^i$ and $X_r^i$ are used as $Q$ in turn. When $\bar{X}_n^i$ serves as $Q$, $X_r^i$ acts as $K$ and $V$ to query features in $X_r^i$ that are similar to $\bar{X}_n^i$. A similar process applies when $X_r^i$ is used as $Q$. Subsequently, at the top branch of the right cross-attention group, since the decoded features $X_{de}^{i+1}$ produced by the deeper decoder layer contains semantic information about glass surfaces, we use $X_{de}^{i+1}$ as $Q$, while the detailed features $\tilde{X}_r^i$ and $\tilde{X}_n^i$ obtained from the first group serve as $K$ and $V$, respectively, to perform cross-attention for locating glass features. The situation at the bottom branch is similar. Finally, we apply the weights $W_r[i]$ and $W_n[i]$ obtained from the IDE module to the results of the right cross-attention

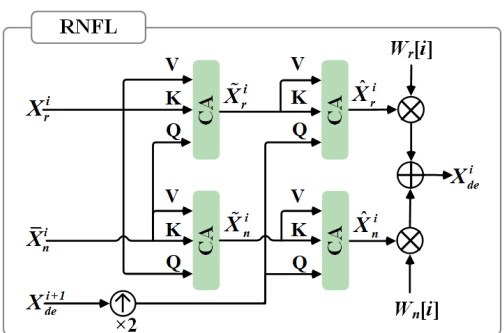

Figure 8: The proposed RNFL module.

group, termed as $\hat{X}_r^i$ and $\hat{X}_n^i$, respectively. The weighted features from the top and bottom branches are then summed to produce the decoded feature $X_{de}^i$. This process can be formulated as:

$$\begin{aligned}
\tilde{X}_r^i &= \text{CA}(\bar{X}_n^i, X_r^i), \quad \tilde{X}_n^i = \text{CA}(X_r^i, \bar{X}_n^i), \\
\hat{X}_r^i &= \text{CA}(X_{de}^{i+1}, \tilde{X}_r^i), \ \hat{X}_n^i = \text{CA}(X_{de}^{i+1}, \tilde{X}_n^i), \\
X_{de}^i &= W_r[i] * \hat{X}_r^i + W_n[i] * \hat{X}_n^i.
\end{aligned} \tag{3}$$

### 4.4 Training Strategy and Loss Functions

As for the loss function, we first train the Retinex Decomposition Stage of our network. We directly use the pre-trained CTDN module (Jiang et al., 2024) for RGB decomposition, and finetune the CTDN module for NIR decomposition using the NIR images of our dataset via the reconstruction loss $L_{rec}$ as follows:

$$L_{rec} = ||I_n - R_n \times L_n||_2. \tag{4}$$

We then train the Encoder and Decoder parts, using the BCE and IoU loss for supervising the glass surface detection and the Dice loss for glass boundary detection. Notably, the CTDN module is frozen in this stage. The BCE loss and IoU loss are adopted to supervise the predicted glass surface mask $P_i$ at $i$-th scale, and the Dice loss (Milletari et al., 2016) for boundary predictions $B_i$. A 3×3 convolution is used as the boundary detection head. The prediction loss function $L_{pred}$ and boundary loss function $L_{bound}$ can be defined as follow:

$$L_{pred} = \sum_{i=0}^{4} (L_{bce}(P_i, P_{gt}) + L_{iou}(P_i, P_{gt})), \tag{5}$$

$$L_{bound} = \sum_{i=1}^{4} L_{dice}(B_i, B_{gt}), \tag{6}$$

where $P_{gt}$ and $B_{gt}$ are the ground truth glass masks and boundary maps, $\text{L}_{bce}(\cdot)$, $\text{L}_{iou}(\cdot)$, and $\text{L}_{dice}(\cdot)$ are the BCE, IoU, and Dice Losses, respectively. $i = 4$ is used to index the glass surface and boundary supervisions at the top of the Encoder. The total loss for the second stage can be defined as follows:

$$L_{stage2} = L_{pred} + \lambda L_{bound}, \tag{7}$$

where $\lambda$ is a hyper-parameter, empirically set to 0.1. Refer to the Supplemental for more training and implementation details.

## 5 Results and Discussion

### 5.1 Nighttime GSD Results

**Quantitative Results.** We compare our method to 17 SOTA methods, including Glass Surface Detection (GSD) methods (Mei et al., 2020; Lin et al., 2021; Fan et al., 2023; Lin et al., 2022; Yan et al., 2025; Lin et al., 2025; Huo et al., 2023; Yan et al., 2024), Mirror Detection (MD) methods (Lin et al., 2020; Xie et al., 2024; He et al., 2023; Mei et al., 2021a), and Salient Object Detection (SOD) methods (Zhou et al., 2023a; 2021; Tu et al., 2021; Cong et al., 2022; Zhou et al., 2023b). All competing methods were retrained on our dataset using the same training/test splits. We used identical training hyperparameters (optimizer, learning rate schedule, epochs) as described in the supplementary materials. Tab. 1 reports the results. While multi-modal methods generally perform better than RGB-based methods, our method achieves the best performance on five evaluation metrics with reasonable computational overheads. We use point prompts generated by the ground-truth mask to obtain the zero-shot predictions of SAM2, and we use the text prompt "glass" to obtain the zero-shot predictions of SAM3.

Tab. 2 reports the results of four best-performing methods (according to Tab. 1, refer to the Supplemental for full results) taking as input the Reflectance components $R_r$ and $R_n$ from the CTDN (Jiang et al., 2024). Since the enhanced images differ from normal daytime images in terms of noise distribution, most methods

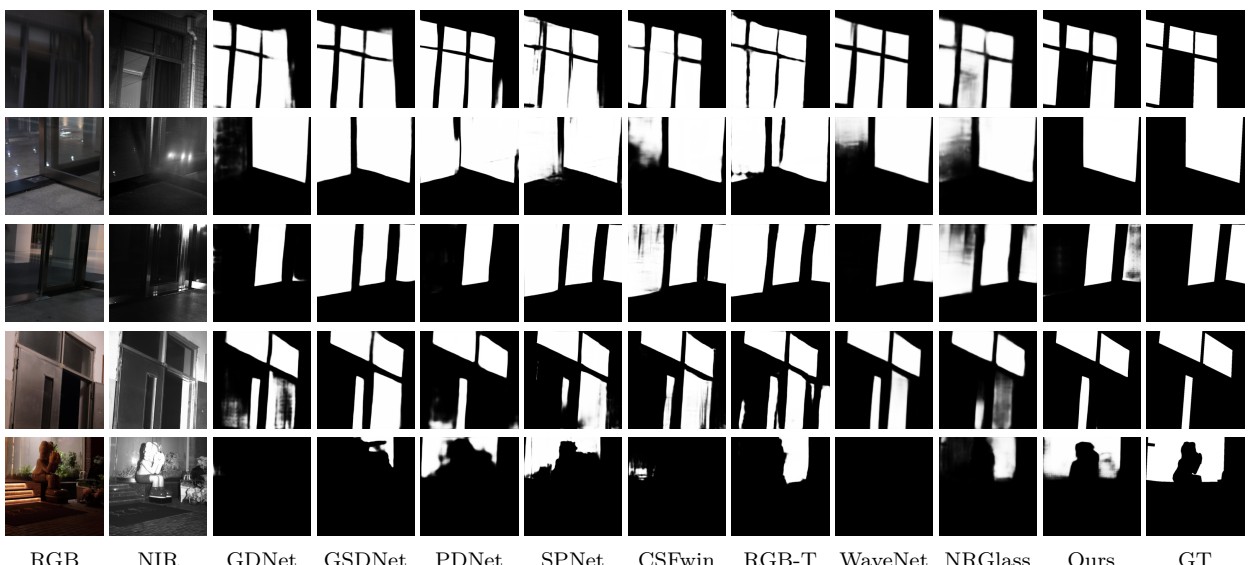

RGB NIR GDNet GSDNet PDNet SPNet CSFwin RGB-T WaveNet NRGlass Ours GT

Figure 9: Visual comparison of our method with 8 competing methods.

Table 1: Quantitative comparison between our method and 17 state-of-the-art methods on our proposed dataset. SAM2 and SAM3 serve as reference baselines. Best detection results are marked in **bold**.

| Modal(s) | Method | Backbone | IoU↑ | $F_\beta$↑ | ACC↑ | MAE↓ | BER↓ | #Params | FLOPs(G) | Time(ms) |
|---|---|---|---|---|---|---|---|---|---|---|
| RGB | SAM2 (Ravi et al., 2024) | ViT | 71.58 | 0.821 | 0.873 | 0.138 | 0.146 | 224.43M | 1620.00 | 84.9 |
| RGB | SAM3 (Carion et al., 2025) | ViT | 60.08 | 0.695 | 0.639 | 0.143 | 0.200 | 840.51M | 5036.94 | 134.9 |
| RGB | GDNet (Mei et al., 2020) | ResNeXt | 78.78 | 0.881 | 0.874 | 0.084 | 0.101 | 201.72M | 207.95 | 229.7 |
| RGB | PMD (Lin et al., 2020) | ResNeXt | 79.50 | 0.871 | 0.893 | 0.079 | 0.094 | 147.66M | 119.26 | 203.7 |
| RGB | GSDNet (Lin et al., 2021) | ResNeXt | 78.87 | 0.876 | 0.891 | 0.085 | 0.100 | 83.72M | 41.27 | 195.2 |
| RGB | GlassSemNet (Lin et al., 2022) | ResNet | 80.12 | 0.886 | 0.907 | 0.088 | 0.093 | 361.33M | 1412.03 | 215.2 |
| RGB | RFENet (Fan et al., 2023) | ResNeXt | 78.85 | 0.881 | 0.882 | 0.083 | 0.099 | 152.65M | 756.91 | 26.3 |
| RGB | HetNet (He et al., 2023) | ResNeXt | 79.33 | 0.870 | 0.899 | 0.082 | 0.093 | 49.59M | 38.73 | 161.1 |
| RGB | CSFwinformer (Xie et al., 2024) | Swin | 82.08 | 0.898 | 0.905 | 0.080 | 0.085 | 230.86M | 188.62 | 187.2 |
| RGB | GhostingNet (Yan et al., 2025) | SwinV2 | 81.37 | 0.889 | 0.910 | 0.077 | 0.091 | 271.53M | 321.70 | 90.7 |
| RGB-D | PDNet (Mei et al., 2021a) | ResNet | 81.47 | 0.891 | 0.910 | 0.074 | 0.084 | 80.54M | 69.85 | 197.9 |
| RGB-D | SPNet (Zhou et al., 2021) | Res2Net | 83.24 | 0.909 | 0.918 | 0.066 | 0.069 | 175.29M | 81.05 | 216.4 |
| RGB-D | CIRNet (Cong et al., 2022) | ResNet | 82.38 | 0.903 | 0.904 | 0.069 | 0.082 | 103.15M | 50.70 | 40.2 |
| RGB-D | RGB-Depth (Lin et al., 2025) | ResNext | 81.41 | 0.895 | 0.909 | 0.075 | 0.084 | 53.28M | 32.13 | 33.8 |
| RGB-T | MIDD (Tu et al., 2021) | ResNet | 77.27 | 0.865 | 0.867 | 0.102 | 0.117 | 79.75M | 169.70 | 92.1 |
| RGB-T | RGBT GSD (Huo et al., 2023) | ResNet | 81.50 | 0.891 | 0.908 | 0.069 | 0.083 | 85.02M | 85.55 | 48.5 |
| RGB-T | PRLNet (Zhou et al., 2023a) | Swin | 83.01 | 0.893 | 0.906 | 0.065 | 0.079 | 570.66M | 277.01 | 144.3 |
| RGB-T | WaveNet (Zhou et al., 2023b) | WaveMLP | 85.66 | 0.922 | 0.925 | 0.057 | 0.066 | 84.88M | 64.02 | 154.3 |
| RGB-NIR | NRGlassNet (Yan et al., 2024) | SwinV2 | 84.54 | 0.917 | 0.914 | 0.066 | 0.073 | 245.30M | 265.35 | 163.2 |
| RGB-NIR | Ours | SwinV2 | **87.98** | **0.934** | **0.936** | **0.047** | **0.055** | 234.88M | 469.98 | 109.2 |
| RGB-NIR | Ours (ablation) | ResNet | 82.63 | 0.901 | 0.905 | 0.070 | 0.081 | 158.42M | 428.81 | 65.1 |
| RGB | Ours* | SwinV2 | 83.51 | 0.906 | 0.835 | 0.065 | 0.075 | 159.60M | 284.27 | 23.6 |

cannot achieve significant improvements. Please refer to the Supplemental for more quantitative comparisons conducted on GDD (Mei et al., 2020) and GSD (Lin et al., 2021) datasets. For further evaluating our method on our nighttime GSD dataset, we also report the results of competing methods directly taking the reflectance components of RGB and NIR images, and the enhanced RGB images produced by LightenDiffusion (Jiang et al., 2024) as input, respectively, in our Supplemental.

**Visual Results.** We compare our method with 8 state-of-the-art methods (Mei et al., 2020; Lin et al., 2021; Huo et al., 2023; Yan et al., 2024; Xie et al., 2024; Mei et al., 2021a; Zhou et al., 2021; 2023b) in Fig. 9. These examples demonstrate that our method can accurately detect glass surfaces under various challenging nighttime conditions (*e.g.*, with opening doors) by exploiting complementary patterns on the glass surfaces between NIR and RGB images, while the competing methods often fail.

Table 2: Comparison with competing methods taking the Reflectance components $R_r$ and $R_n$ as input. Results better than the corresponding ones in Tab. 1 are underlined.

| Modal(s) | Methods | IoU↑ | $F_\beta$↑ | Acc↑ | MAE↓ | BER↓ |
|---|---|---|---|---|---|---|
| RGB | GhostingNet | 81.23 | 0.884 | 0.908 | 0.080 | 0.093 |
| RGB | CSFwinformer | 82.14 | 0.896 | 0.907 | 0.070 | 0.083 |
| RGB-T | WaveNet | 84.59 | 0.912 | 0.902 | 0.066 | 0.077 |
| RGB-NIR | NRGlassNet | 83.84 | 0.911 | 0.910 | 0.070 | 0.078 |
| RGB-NIR | Ours | 87.98 | 0.934 | 0.936 | 0.047 | 0.055 |

Table 3: We test the generalization ability of our method on existing multi-modal (*i.e.*, RGB-NIR (Yan et al., 2024), RGB-Thermal (Huo et al., 2023), and RGB-Depth (Lin et al., 2025)) daytime GSD datasets. All methods are retrained on the corresponding datasets. Refer to the Supplemental for full comparisons. Best results are in **bold**.

| Modal(s) | Methods | Venue | IoU↑ | $F_\beta$↑ | MAE↓ | BER↓ |
|---|---|---|---|---|---|---|
| RGB-NIR | NRGlassNet (Yan et al., 2024) | KBS'24 | **90.03** | **0.955** | **0.033** | **0.036** |
| RGB-NIR | Ours | - | 88.97 | 0.947 | 0.050 | 0.044 |
| RGB-T | RGB-Thermal (Huo et al., 2023) | TIP'23 | 93.80 | 0.965 | **0.027** | **0.040** |
| RGB-T | Ours | - | **94.19** | **0.969** | 0.029 | 0.041 |
| RGB-D | RGB-D GSD (Lin et al., 2025) | AAAI'25 | 74.20 | 0.853 | 0.043 | 0.093 |
| RGB-D | Ours | - | **77.96** | **0.857** | **0.034** | **0.080** |

## 5.2 Daytime GSD Results

We now evaluate the generalization ability of our method on existing daytime GSD datasets (*i.e.*, RGB-NIR (Yan et al., 2024), RGB-Thermal (Huo et al., 2023), and RGB-Depth (Lin et al., 2025) GSD datasets). The results are shown in Tab. 3, where all methods are re-trained on the corresponding datasets. The comparison shows that our method demonstrates reasonable transfer to daytime scenes despite the existing domain discrepancies. [2] Fig. 10 shows some visual comparisons, where we can see that our method can accurately detect the glass surfaces in daytime scenes. Refer to the Supplemental for full comparisons.

Table 4: Ablation study on multi-modal input and our proposed modules.

| Methods | IoU↑ | $F_\beta$↑ | ACC↑ | MAE↓ | BER↓ |
|---|---|---|---|---|---|
| w/o NIR | 84.19 | 0.910 | 0.924 | 0.062 | 0.072 |
| w/o RGB | 82.68 | 0.903 | 0.908 | 0.072 | 0.083 |
| w/o Retinex Dec. | 87.03 | 0.927 | 0.929 | 0.054 | 0.061 |
| w/o RNGE | 87.21 | 0.931 | 0.935 | 0.051 | 0.059 |
| w/o IDE | 87.35 | **0.935** | 0.933 | 0.048 | 0.057 |
| w/o RNFL | 87.09 | 0.915 | 0.931 | 0.049 | 0.060 |
| w/o $X_r^3$ | 87.34 | 0.932 | 0.936 | 0.049 | 0.058 |
| w/o $X_r^i$ | 87.42 | 0.933 | 0.931 | 0.049 | 0.059 |
| w/o subtraction | 87.67 | 0.933 | 0.936 | 0.048 | 0.056 |
| Ours | **87.98** | 0.934 | **0.936** | **0.047** | **0.055** |

## 5.3 Ablation Results

We report ablation results on the proposed dataset in Tab. 4. The first two rows show that using either single modality (*i.e.*, "w/o NIR" and "w/o RGB") significantly decreases the performance. The 3rd to 6th rows show the results when we individually ablate the Retinex decomposition, RNGE, IDE, and RNFL modules. Though the results indicate that the contributions of individual modules are consistent but relatively small, the proposed modules are designed to be complementary rather than purely additive, with each addressing a

---

[2]Note that the NIR images in the RGB-NIR dataset (Yan et al., 2024) are captured by placing an NIR filter over a DSLR camera lens, which differs from our setup.

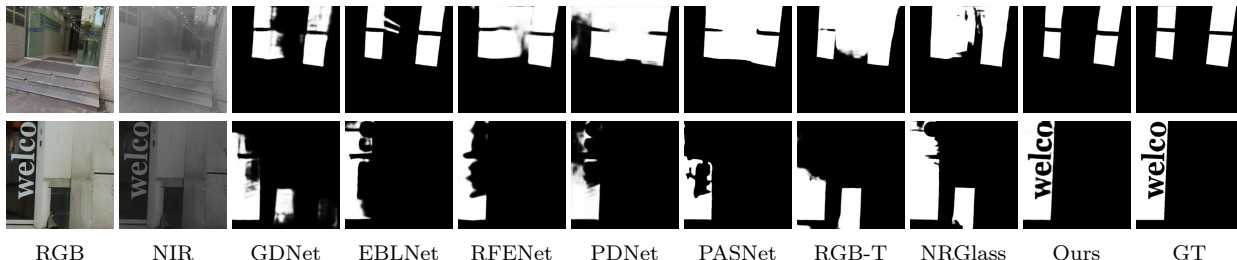

RGB    NIR    GDNet    EBLNet    RFENet    PDNet    PASNet    RGB-T    NRGlass    Ours    GT

Figure 10: Comparison of our method with competing methods on daytime scenes.

different aspect of the problem: RNGE enhances informative reflectance cues through cross-modal differences, IDE provides illumination-aware guidance based on Retinex decomposition, and RNFL progressively fuses multi-scale features under illumination guidance. The 7th to 9th rows show the ablation results obtained by removing specific input features and the subtraction operation from our RNGE model. Moreover, we replace the backbone with ResNet-50, termed as "Ours(ablation)", and ablate our network to a single-branch network (no NIR input, no NIR feature encoder), termed as "Ours*", as shown in Tab. 1. The results demonstrate that the "SwinV2" is more effective and our complete model obtains the best performance. Refer to the Appendix for more implementation details, results, and analysis of ablations.

## 5.4 Limitations

Despite the effectiveness of our method, it does have some limitations. When RGB (visible) information is absent, the near-infrared images provide insufficient cues to support accurate glass detection, because the NIR images are presented as gray-scale and lack certain color and texture information. The first scene of Fig. 11 shows that our method over-detects the bottom-right non-glass region (an open window) as the glass region, as this region contains similar patterns to the upper-right glass region in both the RGB and NIR modalities. The second scene of Fig. 11 shows that if a glass region is too dark for the retinal decomposition method to obtain the desired reflection component, our method may not detect it.

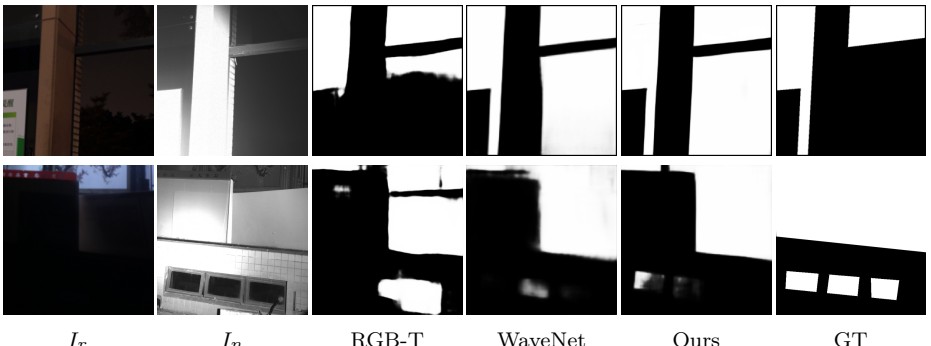

$I_r$      $I_n$      RGB-T      WaveNet      Ours      GT

Figure 11: Failure cases of our proposed method.

## 6 Conclusion

We have proposed a novel method for nighttime glass surface detection by modeling the complementary patterns of glass surface regions between RGB and NIR image pairs. Our network has a novel RNGE module for RGB-to-NIR guiding feature enhancement and a novel RNFL module for glass surface detection based on the guided multimodal feature aggregation. We have also constructed the first large-scale nighttime glass surface detection dataset. Extensive evaluations show that our proposed method outperforms SOTA methods in nighttime scenes while maintaining competitive performance in daytime conditions.

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

## A  Appendix and Supplementary Materials

In this appendix, we provide implementation details of our method, more analysis of the impact of pre-processing operations on input data. We also provide more qualitative visual comparisons between existing state-of-the-art methods from relevant fields and our model. Finally, we analyzed the impact of different modal inputs on nighttime glass detection problems and also examined the practical role of Retinex decomposition.

### A.1  Implementation Details

Our network is implemented using PyTorch on an Nvidia RTX 4090 GPU. Swin Transformer V2 (Liu et al., 2022b) pre-trained on ImageNet-1K is adopted as the backbone of our network. The resolution of the image input is set to $384 \times 384$. For data augmentation, we follow the previous work (Lin et al., 2021) to use random cropping, random rotation, and random horizontal flip. We use the AdamW (Loshchilov & Hutter, 2019) optimizer, while the initial learning rate is set to $1 \times 10^{-5}$, and the batch size is set to 2. We train our model with 100 epochs, which takes about 10 hours.

Specifically, our training pipeline consists of two separate stages, which follow a consistent training protocol.

(1) Stage 1 (Retinex decomposition and CTDN fine-tuning): We initialize the CTDN module with pre-trained weights and fine-tune it for NIR decomposition using the reconstruction loss $L_{rec}$. During this stage, only the CTDN parameters are updated, while the rest of the network remains inactive.

(2) Stage 2 (Glass detection network training): After Stage 1, the CTDN module is frozen, and we train the encoder–decoder network for glass surface and boundary detection using $L_{stage2}$.

### A.2  Supplemental Quantitative Comparison

Moreover, Tab. 5 reports the evaluation results of competing methods when using the reflectance components of RGB and NIR images (from our nighttime GSD dataset), decomposed by CTDN (Jiang et al., 2024), as input. In other words, the reflectance components of RGB ($R_r$) and NIR ($R_n$) images takes place of the original RGB ($I_r$) and NIR ($I_n$) images. Tab. 6 reports the results of the competing methods taking as input the enhanced RGB images (from our nighttime GSD dataset), produced by the SOTA low-light enhancement method (Jiang et al., 2024). The results in Tab. 5 and 6 indicate that, even with Retinex decomposition or low-light enhancement, existing methods fail to achieve consistent performance improvements, largely due to the distribution shift between nighttime and daytime images. Our model not only uses the RNGE module to explore the differences and complementary information between the two modalities, but also takes the

Table 5: Study on competing methods directly taking Reflectance component $R_r$ (and $R_n$) as input. Each value better than its corresponding value shown in Tab. 1 of our paper is marked in underlined.

| Methods | IoU↑ | $F_\beta$↑ | Acc↑ | MAE↓ | BER↓ |
|---------|------|-----------|------|------|------|
| GDNet (Mei et al., 2020) | 78.22 | 0.872 | 0.878 | 0.085 | 0.103 |
| GSDNet (Lin et al., 2021) | 77.98 | 0.876 | 0.873 | 0.085 | 0.105 |
| GlassSemNet (Lin et al., 2022) | 79.05 | 0.879 | 0.887 | 0.088 | 0.097 |
| RFENet (Fan et al., 2023) | 78.53 | 0.872 | 0.887 | 0.086 | 0.102 |
| GhostingNet (Yan et al., 2025) | 81.23 | 0.884 | 0.908 | 0.080 | 0.093 |
| PMD (Lin et al., 2020) | 78.89 | 0.866 | 0.889 | 0.083 | 0.097 |
| HetNet (He et al., 2023) | 78.59 | 0.863 | 0.895 | 0.086 | 0.096 |
| CSFwinformer (Xie et al., 2024) | 82.14 | 0.896 | 0.907 | 0.070 | 0.083 |
| PDNet (Mei et al., 2021a) | 80.77 | 0.889 | 0.903 | 0.077 | 0.086 |
| SPNet (Zhou et al., 2021) | 82.53 | 0.904 | 0.907 | 0.072 | 0.078 |
| CIRNet (Cong et al., 2022) | 81.34 | 0.897 | 0.898 | 0.076 | 0.086 |
| RGB-Depth (Lin et al., 2025) | 80.94 | 0.887 | 0.903 | 0.078 | 0.086 |
| MIDD (Tu et al., 2021) | 76.96 | 0.863 | 0.866 | 0.105 | 0.118 |
| RGB-Thermal (Huo et al., 2023) | 79.09 | 0.877 | 0.891 | 0.084 | 0.091 |
| PRLNet (Zhou et al., 2023a) | 82.37 | 0.887 | 0.901 | 0.067 | 0.083 |
| WaveNet (Zhou et al., 2023b) | 84.59 | 0.912 | 0.902 | 0.066 | 0.077 |
| NRGlassNet (Yan et al., 2024) | 83.84 | 0.911 | 0.910 | 0.070 | 0.078 |
| Ours | 87.98 | 0.934 | 0.936 | 0.047 | 0.055 |

Table 6: Comparisons with competing methods taking the enhanced RGB images produced by LightenDiffusion (Jiang et al., 2024) as input. Results better than the corresponding ones in Tab. 1 of our paper are underlined.

| Methods | IoU↑ | $F_\beta$↑ | ACC↑ | MAE↓ | BER↓ |
|---------|------|-----------|------|------|------|
| GDNet (Mei et al., 2020) | 78.82 | 0.878 | 0.880 | 0.082 | 0.100 |
| GSDNet (Lin et al., 2021) | 78.92 | 0.878 | 0.880 | 0.081 | 0.100 |
| GlassSemNet (Lin et al., 2022) | 80.20 | 0.885 | 0.900 | 0.080 | 0.094 |
| RFENet (Fan et al., 2023) | 79.22 | 0.882 | 0.878 | 0.080 | 0.099 |
| GhostingNet (Yan et al., 2025) | 81.29 | 0.887 | 0.911 | 0.079 | 0.092 |
| PMD (Lin et al., 2020) | 79.54 | 0.870 | 0.893 | 0.080 | 0.094 |
| HetNet (He et al., 2023) | 78.82 | 0.869 | 0.887 | 0.083 | 0.097 |
| CSFwinformer (Xie et al., 2024) | 82.75 | 0.899 | 0.908 | 0.066 | 0.079 |
| PDNet (Zhou et al., 2021) | 81.16 | 0.892 | 0.910 | 0.076 | 0.085 |
| SPNet (Zhou et al., 2021) | 82.88 | 0.896 | 0.914 | 0.075 | 0.071 |
| CIRNet (Cong et al., 2022) | 82.43 | 0.903 | 0.900 | 0.067 | 0.082 |
| RGB-Depth (Lin et al., 2025) | 81.46 | 0.891 | 0.911 | 0.073 | 0.086 |
| MIDD (Tu et al., 2021) | 77.53 | 0.867 | 0.865 | 0.098 | 0.102 |
| RGB-Thermal (Huo et al., 2023) | 81.70 | 0.896 | 0.904 | 0.080 | 0.087 |
| PRLNet (Zhou et al., 2023a) | 83.07 | 0.891 | 0.909 | 0.068 | 0.080 |
| WaveNet (Zhou et al., 2023b) | 85.73 | 0.921 | 0.927 | 0.055 | 0.067 |
| NRGlassNet (Yan et al., 2024) | 84.47 | 0.913 | 0.916 | 0.067 | 0.075 |
| Ours | **87.98** | **0.934** | **0.936** | **0.047** | **0.055** |

unbalanced nighttime lighting conditions into account through the IDE and RNFL modules. Therefore, our model can handle more complex nighttime situations rather than simply combining two tasks. We provide several examples of enhanced images in Fig. 16 and Fig. 17.

We conducted additional experiments on two single-image glass surface detection benchmarks GDD (Mei et al., 2020) and GSD (Lin et al., 2021) to validate the robustness of our model. Following the experimental observations, we adopt two identical RGB images as inputs to produce the final results, while keeping the network architecture unchanged. As shown in the Tab. 7, our model performs slightly worse than the latest glass surface detection methods based on visual foundation models (GlassWizard (Li et al., 2025) and

MSNet (Cheng et al., 2026)). Nevertheless, the results suggest that our model can competitively handle daytime scenarios.

Table 7: Quantitative comparison between our method and 8 state-of-the-art methods on the benchmark datasets GDD and GSD. Best detection results are marked in **bold**.

| Methods | Venue | GDD | | | | GSD | | | |
|---|---|---|---|---|---|---|---|---|---|
| | | IoU↑ | $F_\beta$↑ | MAE↓ | BER↓ | IoU↑ | $F_\beta$↑ | MAE↓ | BER↓ |
| GDNet | CVPR'20 | 87.63 | 0.937 | 0.063 | 0.056 | 79.01 | 0.869 | 0.069 | 0.077 |
| GSDNet | CVPR'21 | 88.07 | 0.932 | 0.059 | 0.057 | 83.64 | 0.903 | 0.055 | 0.061 |
| EBLNet | ICCV'21 | 88.16 | 0.939 | 0.059 | 0.056 | 85.04 | 0.916 | 0.053 | 0.064 |
| RFENet | IJVAI'23 | 88.72 | 0.940 | 0.055 | 0.054 | 86.50 | 0.931 | 0.048 | 0.062 |
| VBNet | TII'24 | 90.58 | 0.944 | 0.048 | 0.047 | 85.90 | 0.915 | 0.043 | 0.054 |
| GhostingNet | TPAMI'25 | 89.30 | 0.943 | 0.054 | 0.051 | 83.77 | 0.904 | 0.055 | 0.061 |
| GlassWizard | ICCV'25 | **92.10** | **0.961** | **0.041** | **0.039** | **89.10** | **0.942** | **0.035** | **0.041** |
| MSNet | AAAI'26 | 91.50 | 0.955 | 0.043 | 0.041 | 87.80 | 0.916 | 0.042 | 0.047 |
| Ours | - | 91.27 | 0.946 | 0.044 | 0.045 | 87.91 | 0.921 | 0.039 | 0.047 |

## A.3 Supplemental Qualitative Comparison

Fig. 12, 13 and 14 show more qualitative comparison results produced by our method and 8 SOTA competing methods (Mei et al., 2020; Lin et al., 2021; Huo et al., 2023; Yan et al., 2024; Xie et al., 2024; Mei et al., 2021a; Zhou et al., 2021; 2023b).

Fig. 12 shows that smooth and flat non-glass surfaces can easily lead to misjudgment in glass detection methods due to the reflection of visible light, mistakenly identifying it as a glass area. Our method utilizes the characteristic of similar reflection phenomena on non-glass surfaces and significant differences in reflection phenomena on glass surfaces in two modalities to distinguish glass regions.

Fig. 13 shows that open doors and windows have the same edge features as closed glass doors and windows, leading to the failure of edge-dependent detection methods. Our method focuses on the differences between the two modes in the door and window area, so it can accurately distinguish between open doors and windows and closed glass doors and windows.

Fig. 14 shows that complex/curved glass boundaries (corresponding to partially obscured glass surfaces) pose challenges to glass surface detection methods. Ordinary glass surfaces typically exhibit regular geometric shapes, so curved glass boundaries can contradict this common assumption.

Fig. 15 shows additional qualitative results on the daytime NIR dataset (Yan et al., 2024). These qualitative results demonstrate that our method performs better than the competing methods.

Fig. 16 shows low-light enhanced image produced by LightenDiffusion (Jiang et al., 2024). These images are from our nighttime GSD dataset.

Fig. 17 shows the reflectance components ($R_r$ and $R_n$), the illumination components ($L_r$ and $L_n$) decomposed by the CTDN Module of (Jiang et al., 2024), the detected result produced by our method and the ground truth mask.

## A.4 Discussion on Additional Modalities for Nighttime GSD

Due to insufficient lighting, the scattering and reflection intensity of glass is weak, making nighttime glass surface detection challenging for networks that only use RGB images. Introducing active visible lighting (such as RGB with a flash) can enhance specular reflections on the glass surface; however, this may also introduce strong glare and color distortion.

Alternative spectral modalities provide additional cues. Short-wave infrared (SWIR) and thermal infrared imaging can also be effective for nighttime glass surface detection; however, achieving high spatial resolution or satisfactory image quality with these modalities typically requires relatively expensive sensing hardware.

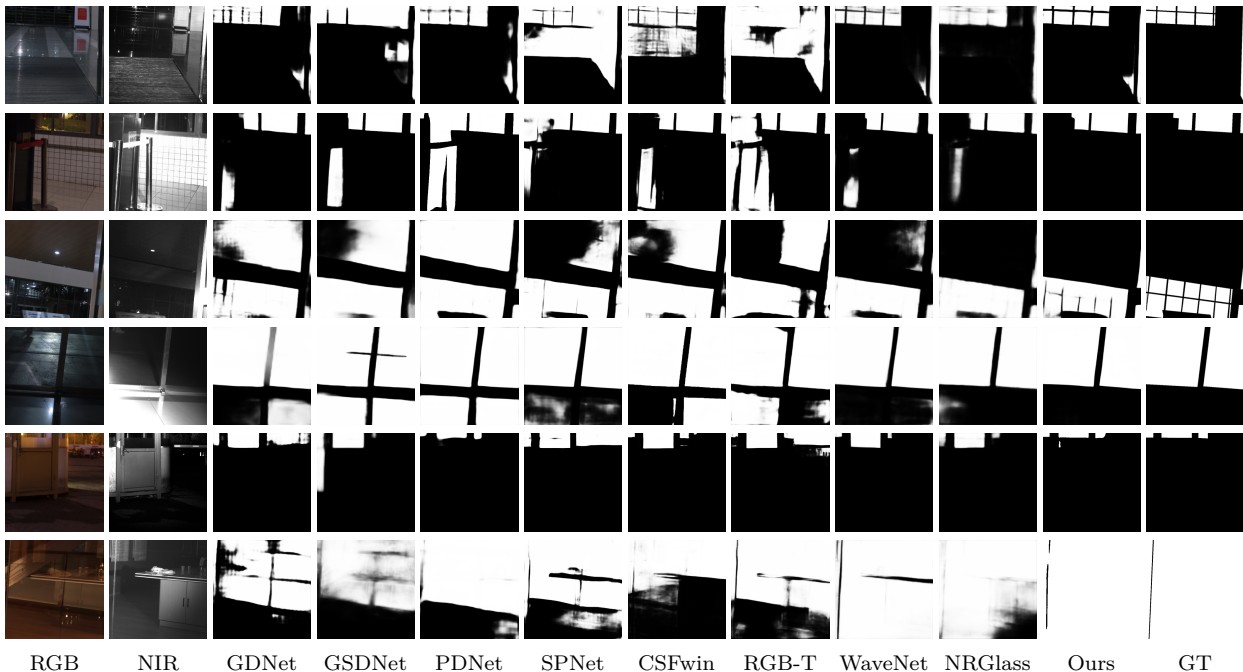

Figure 12: More experiment results on challenging scenes with smooth non-glass surfaces.

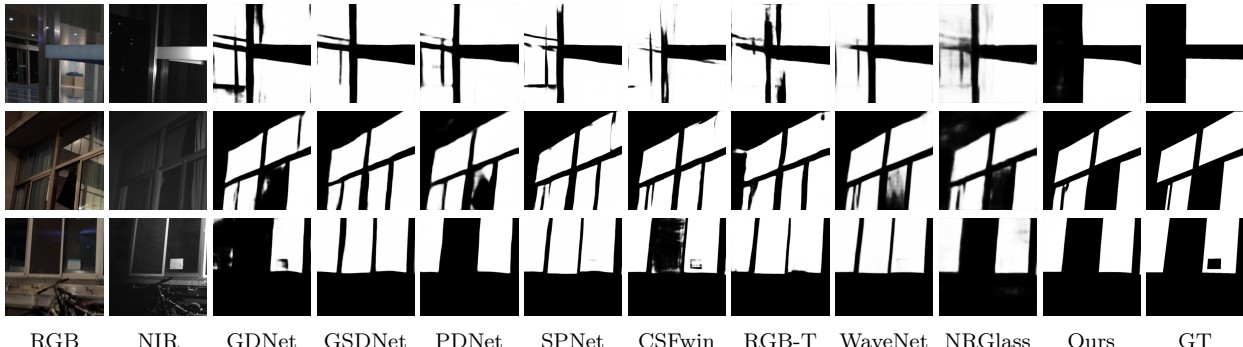

Figure 13: More experiment results on challenging scenes with opened windows and doors.

Depth sensing is another commonly used modality for glass-related perception tasks; however, its effectiveness will be limited due to noisy measurements.

Near-infrared (NIR) imaging offers a favorable trade-off in terms of image quality, sensor cost, and deployment flexibility. It enables active illumination with relatively low-cost sensors and shares similar image formation principles with visible-light imaging, while exhibiting more stable reflection behavior at glass surfaces. These physical considerations motivate our choice of an RGB–NIR sensing configuration for nighttime glass surface detection.

## A.5 Study on different Multi-modal Inputs

We have also conducted an experiment to study the different combinations of multimodal image data (such as RGB, Depth, NIR and thermal images) for glass surface detection. The hybrid imaging systems we used to capture multimodal image data are shown in Fig. 22. As shown in Fig. 18, 19 and 21, we captured RGB images, NIR images, thermal images, and depth maps of the target scenes. Then, we compare our method with the competing multi-modal methods including RGB-D based method (Lin et al., 2025), RGB-T based

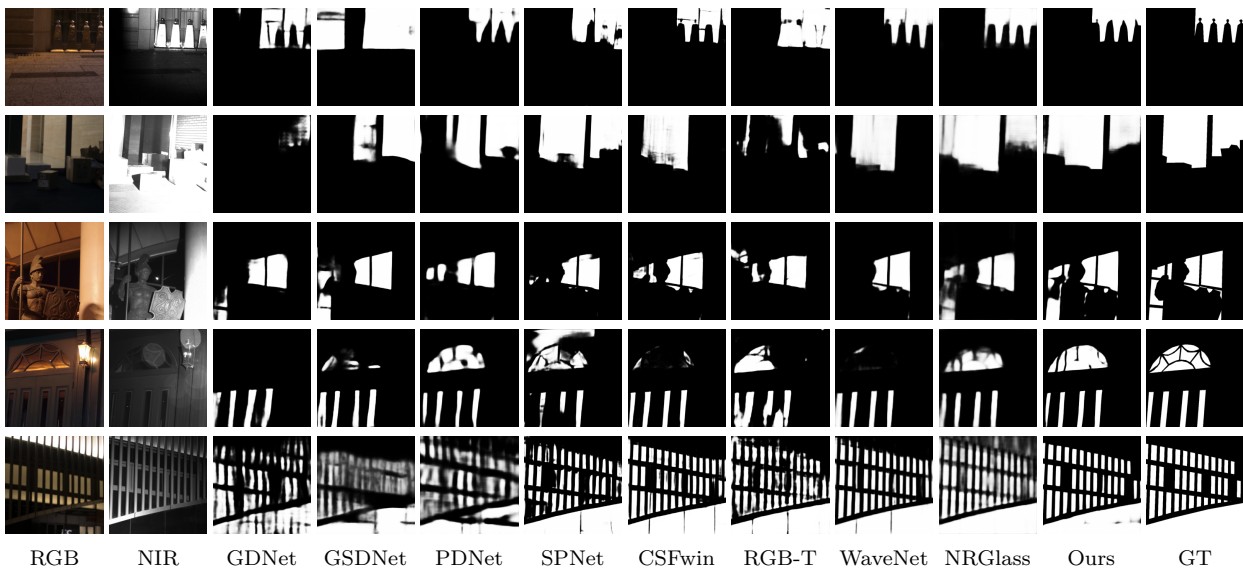

RGB   NIR   GDNet   GSDNet   PDNet   SPNet   CSFwin   RGB-T   WaveNet   NRGlass   Ours   GT

Figure 14: More experiment results on challenging scenes with complex boundaries.

RGB   NIR   GDNet   EBLNet   RFENet   PDNet   PASNet   RGB-T   NRGlass   Ours   GT

Figure 15: Comparison of our method with competing methods on daytime scenes.

Figure 16: Visualization of original images from our dataset (top row) and enhanced images (bottom row) produced by LightenDiffusion (Jiang et al., 2024).

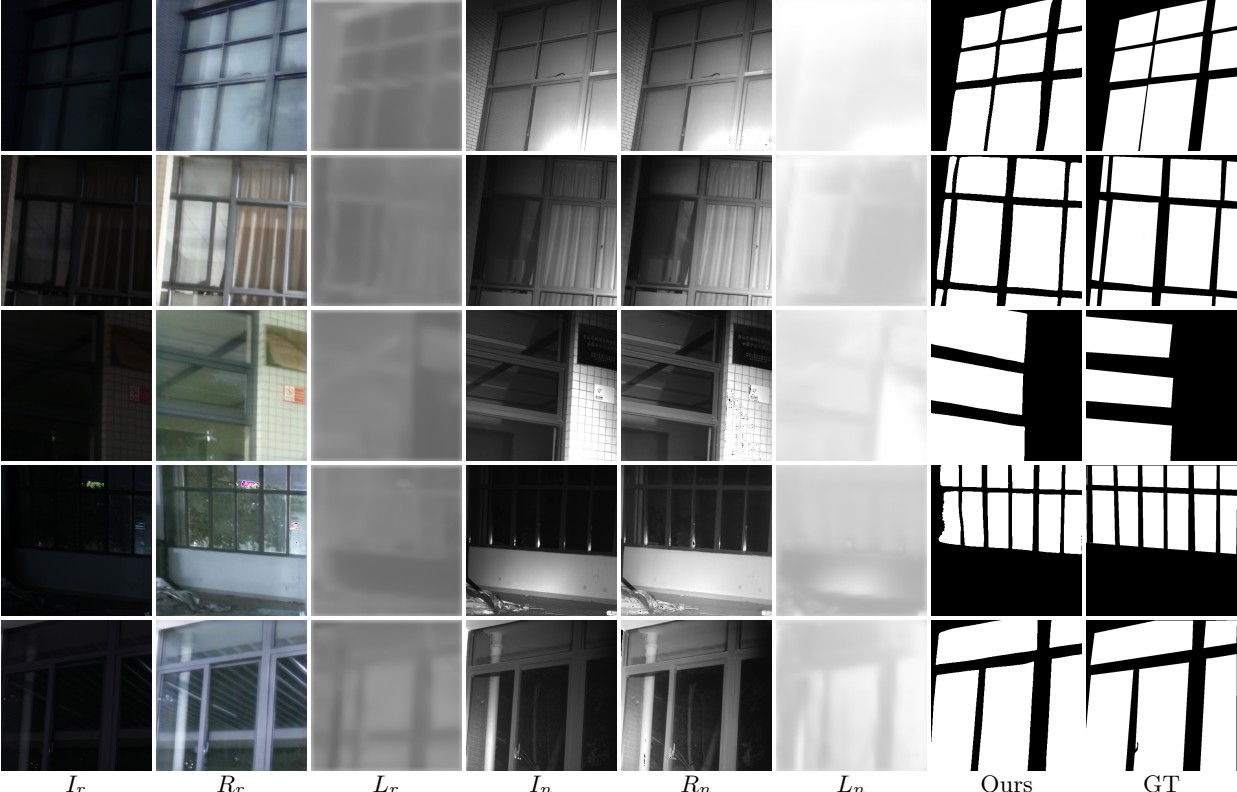

$$I_r \qquad R_r \qquad L_r \qquad I_n \qquad R_n \qquad L_n \qquad \text{Ours} \qquad \text{GT}$$

Figure 17: Retinex decomposition results for dark conditions. For each nighttime scene, from left to right, we show the reflectance and illumination components ($R_r$ and $L_r$) of the RGB image, the corresponding components ($R_n$ and $L_n$) of the NIR image, the result of our method and the ground truth.

method(Huo et al., 2023) on the captured multimodal image data. These images are not included in our dataset.

The $1st$ scene of Fig. 21 shows that, in low-light scenes, due to the lack of the sun as a thermal radiation source, the intensity difference between different regions in the thermal images is very small. The $2nd$ scene of Fig. 21 shows that thermal image is susceptible to interference from external strong thermal radiation sources, such as the air conditioner outdoor unit reflected on glass surface in the thermal image. The $3rd$ scene of Fig. 21 shows that depth values of the glass door is incorrect, which are close to the depth values of the outdoor scene. Thus, RGB-D based method (Lin et al., 2025) cannot distinguish glass door and open glass door.

These examples demonstrate that our captured RGB-NIR image pair can provide more valuable information than RGB-D and RGB-T image pairs for glass surface detection, and our method outperforms those RGB-D and RGB-T based methods in various challenging nighttime scenes.

### A.6 Study on misalignment Inputs

In our framework, the RNGE module adopts subtraction of feature maps to capture complementary cues, while the RNFL module performs feature addition for fusion. Both operations inherently assume well-aligned features to ensure stable and accurate fusion. To evaluate the robustness of our model under practical conditions, we conducted experiments with synthetically misaligned RGB-NIR inputs. Specifically, we introduced random spatial shifts to the NIR images to simulate imperfect camera calibration and synchronization. As the offset increased, performance gradually degraded: a small misalignment (about 5 pixels) led to an IoU drop of only 1.1, while a larger shift (about 10 pixels) caused a 2.4 decrease in IoU, as shown in Tab. 8. These results indicate that our model is reasonably tolerant to moderate spatial misalignment, which is a common issue in

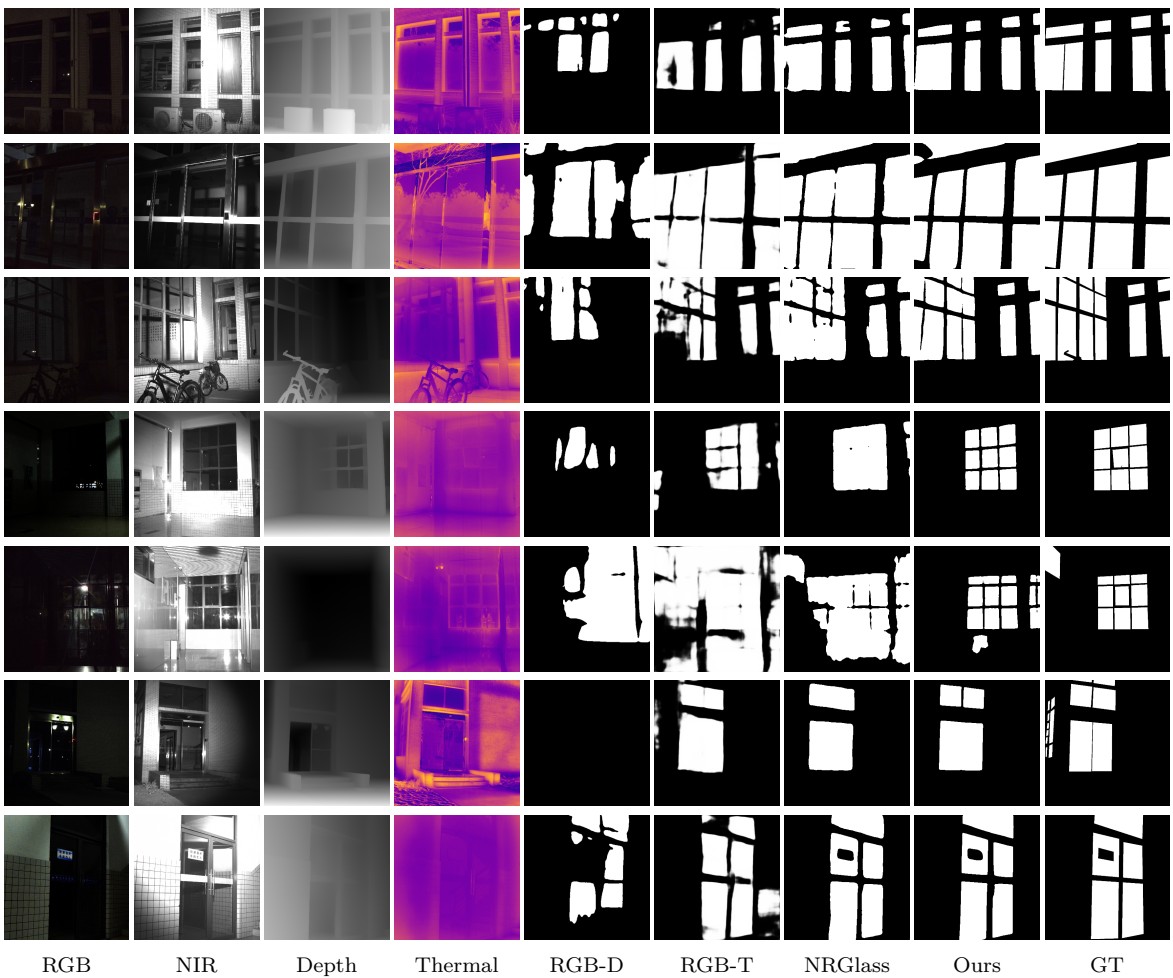

Figure 18: Study on different multi-modal inputs.

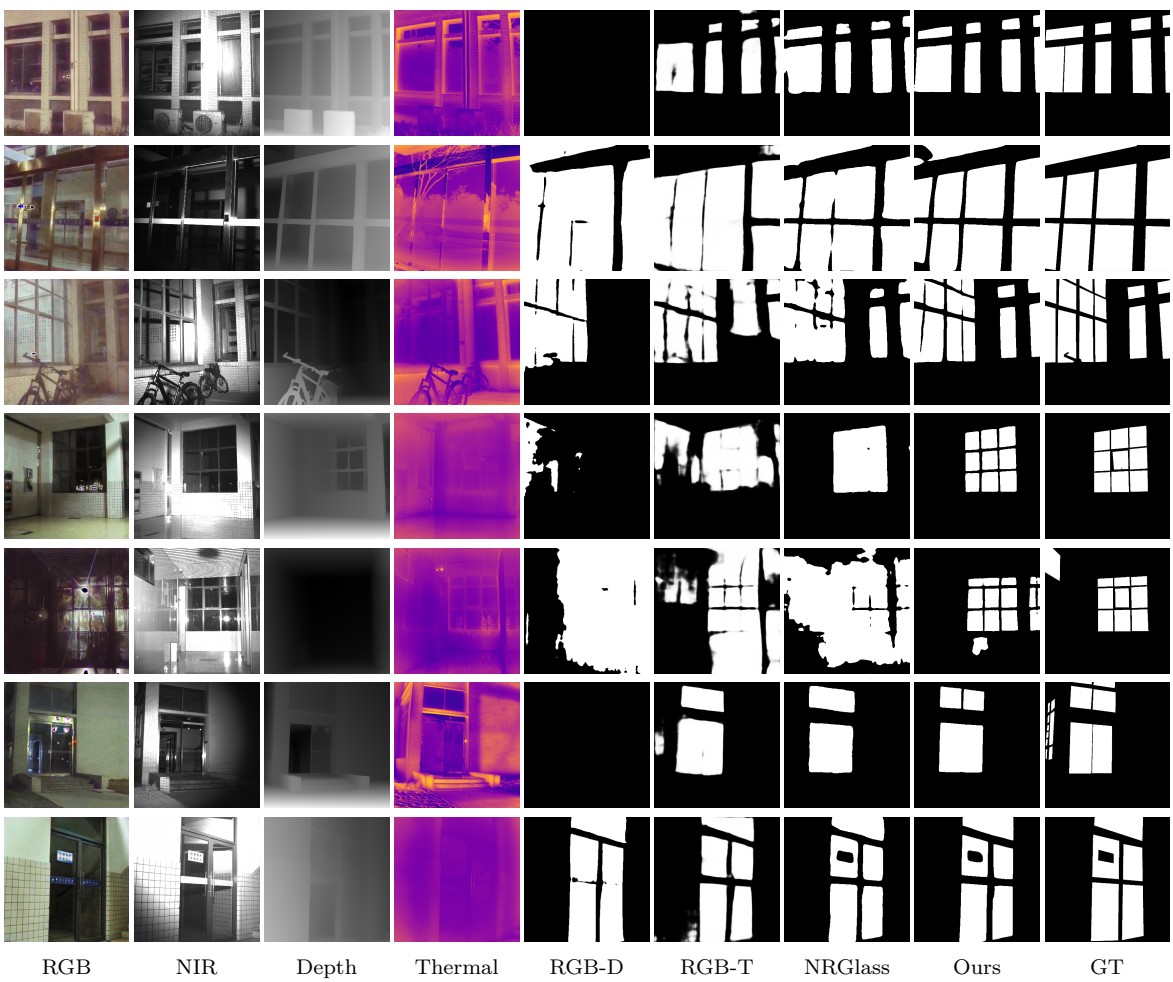

RGB          NIR          Depth          Thermal          RGB-D          RGB-T          NRGlass          Ours          GT

Figure 19: Use the Reflectance component as input.

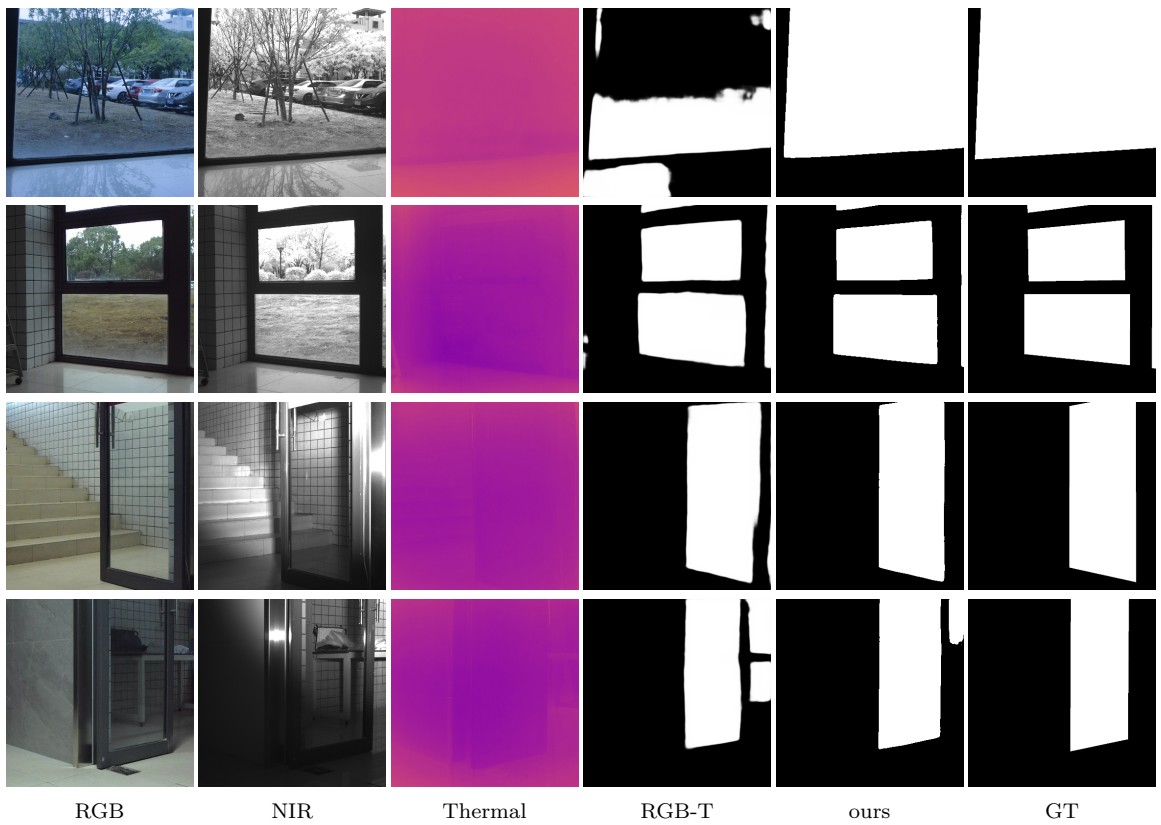

RGB            NIR            Thermal            RGB-T            ours            GT

Figure 20: Some indoor low heat radiation difference scenes.

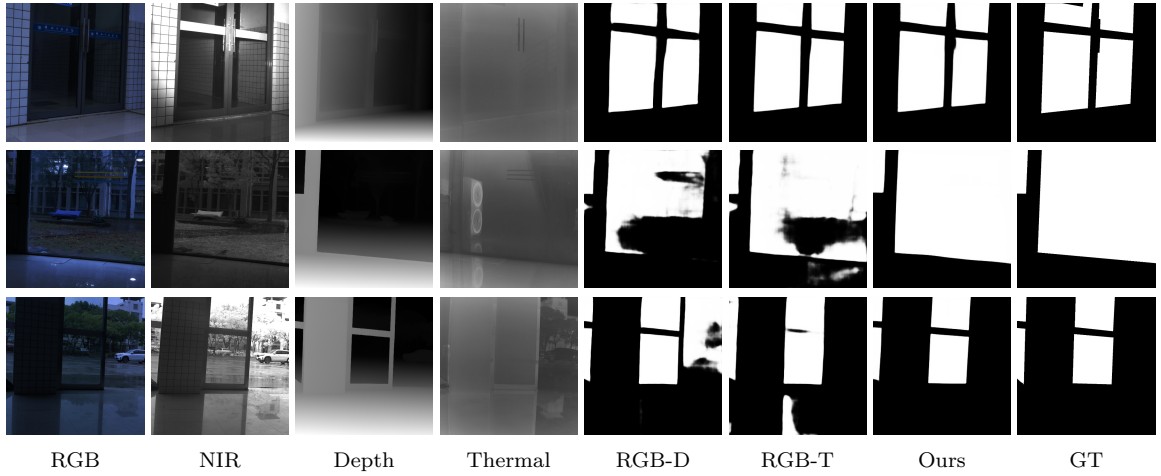

RGB       NIR       Depth       Thermal       RGB-D       RGB-T       Ours       GT

Figure 21: Study on different multi-modal inputs in low-light environment.

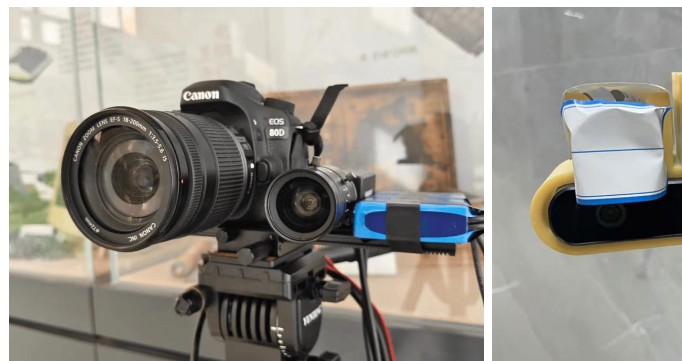

(a) Our designed hybrid imaging system.  (b) RGB-T and RGB-D imaging system.

Figure 22: (a) shows our designed hybrid imaging system for capturing RGB-NIR image pairs. It consists of a DSLR camera and an NIR camera. (b) shows the hybrid imaging system consisting of a thermal infrared camera and a stereo camera we used for capturing thermal images and depth maps.

real-world dual-camera systems, and can still maintain acceptable performance under imperfect alignment. Notably, misalignment primarily introduces boundary perturbations in the predicted masks, as the model outputs a single fused mask without distinguishing which modality contributes to the detection. As shown in Fig. 23, when the RGB image is relatively dark, our model tends to output the glass region in the NIR image.

Table 8: The impact of misaligned images.

| pixel range | IoU↑ | $F_\beta$↑ | MAE↓ | BER↓ | ACC↑ |
|---|---|---|---|---|---|
| 0-0 pixel | 87.98 | 0.934 | 0.047 | 0.055 | 0.936 |
| 0-5 pixel | 86.88 | 0.928 | 0.051 | 0.060 | 0.929 |
| 0-10 pixel | 85.58 | 0.923 | 0.054 | 0.067 | 0.918 |

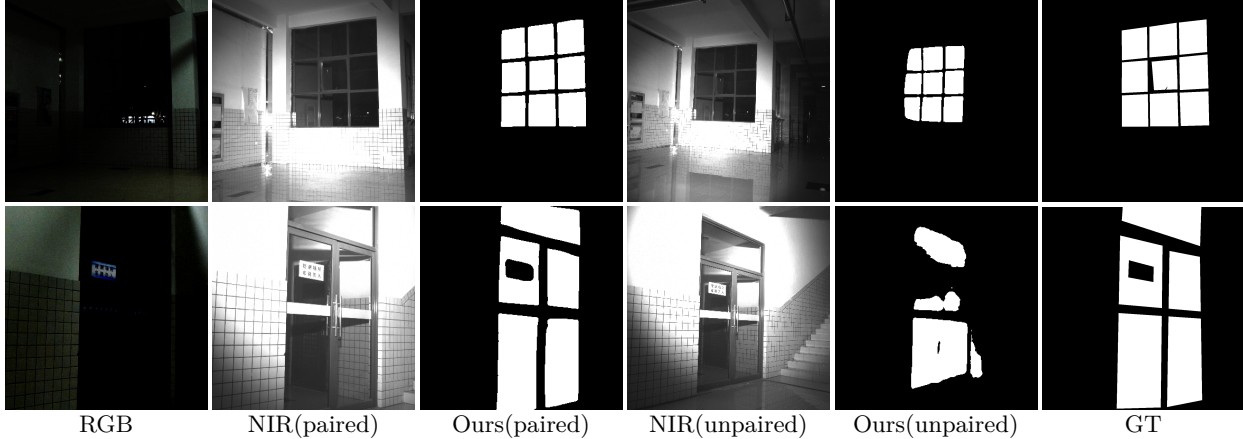

RGB          NIR(paired)     Ours(paired)     NIR(unpaired)     Ours(unpaired)          GT

Figure 23: Qualitative results of our model tested on unaligned images.

## A.7 Study on glass object detection

The glass object detection (GOD) task substantially differs from the glass surface detection (GSD) task, due to the distinct properties of their respective targets. Specifically, GOD focuses on detecting glass objects characterized by their shapes and boundaries, whereas GSD emphasizes glass surfaces, which are defined by reflection and transmission phenomena.

We evaluate our method on the Trans-10K dataset (Xie et al., 2020), as shown in Tab. **??**. This dataset contains two subsets of transparent objects: (1) Transparent things, referring to small-scale curved objects such as cups, bottles, and glasses; and (2) Transparent stuff, referring to large-scale surfaces such as windows, glass walls, and doors.

Table 9: Quantitative comparison between our method and TransLab(Xie et al., 2020) on the dataset Trans-10k. Best detection results are marked in **bold**.

| Method | MAE ↓ | ACC ↑ | IoU ↑ | | BER ↓ | |
|---|---|---|---|---|---|---|
| | | | Things | Stuff | Things | Stuff |
| TransLab | 0.063 | 92.69 | 90.87 | 84.39 | 3.63 | 7.28 |
| Ours | **0.022** | **96.00** | **94.37** | **89.47** | **1.80** | **4.70** |

## A.8 Study on Different Materials of Glass Surface

We conducted experiments on multiple glass variants, including tinted glass (which partially blocks IR), frosted glass (which scatters light), and coated glass (using low-E coatings). The qualitative results are presented in Fig. 24. These results demonstrate that our model maintains accurate glass detection across all tested glass types, despite their different physical properties and NIR responses. Our model does not rely on a fixed NIR reflectance threshold; instead, it learns the relative spectral discrepancy between RGB and NIR modalities. The IDE module dynamically adjusts the fusion weights between RGB and NIR modalities. This allows the system to adaptively prioritize the more informative modality, thereby mitigating sensitivity to specific hardware or material-induced intensity variations.

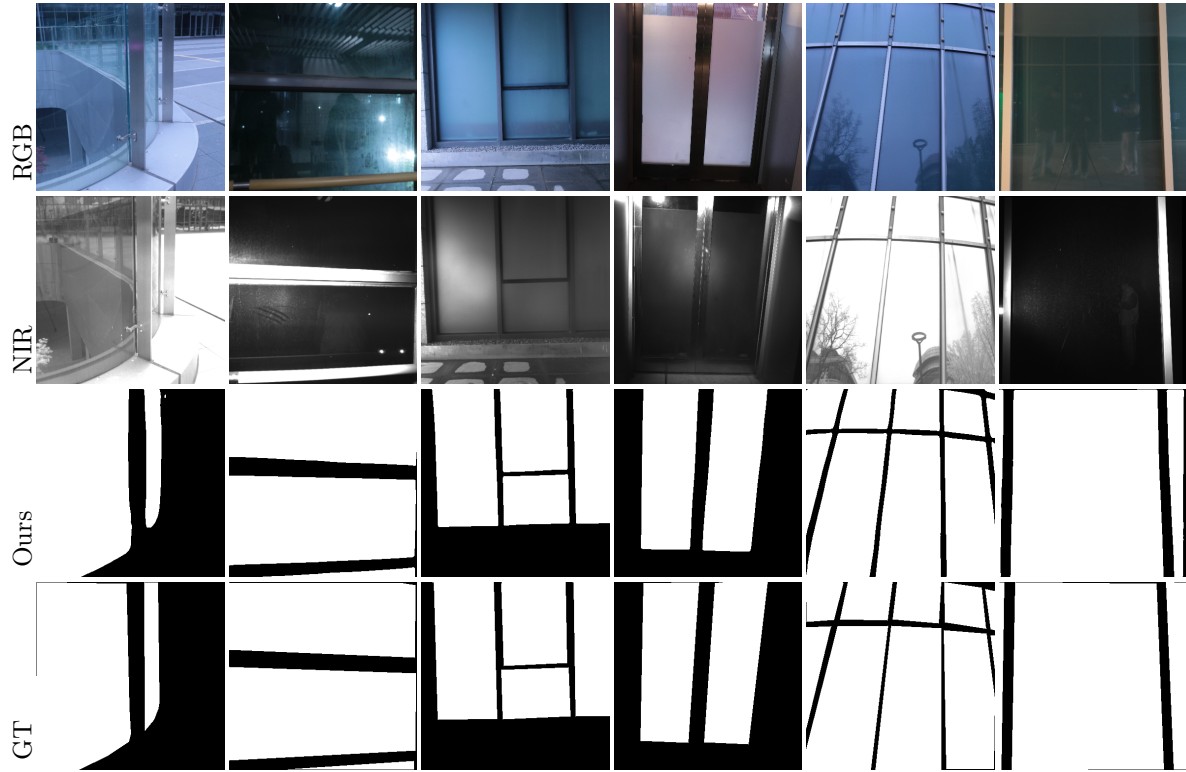

Figure 24: Qualitative results of our method on different types of glass. From left to right: the 1*st* and 2*nd* scenes feature tinted glass; the 3*rd* and 4*th* scenes feature frosted glass; the 5*th* and 6*th* scenes feature coated glass.

### A.9 Study on Curved Glass Surfaces

Compared to flat glass surfaces, curved glass surfaces exhibit several unique visual properties, such as curved boundaries and distorted reflections. Since most of our dataset consists of smooth and flat glass, this introduces a data bias. Despite this bias, our method maintains reasonable performance across different scenes in our evaluation, as illustrated in Fig. 25. However, our model may under-detect or over-detect on curved glass surfaces, as shown in the $5th$ to $7th$ scenes of Fig. 25. This is largely due to the dataset bias, as most of the training samples consist of smooth and planar glass. To further improve generalization, expanding the dataset to include more diverse glass types (e.g., curved, fractured, or heavily textured surfaces) and spatial distributions is an important direction for future work.

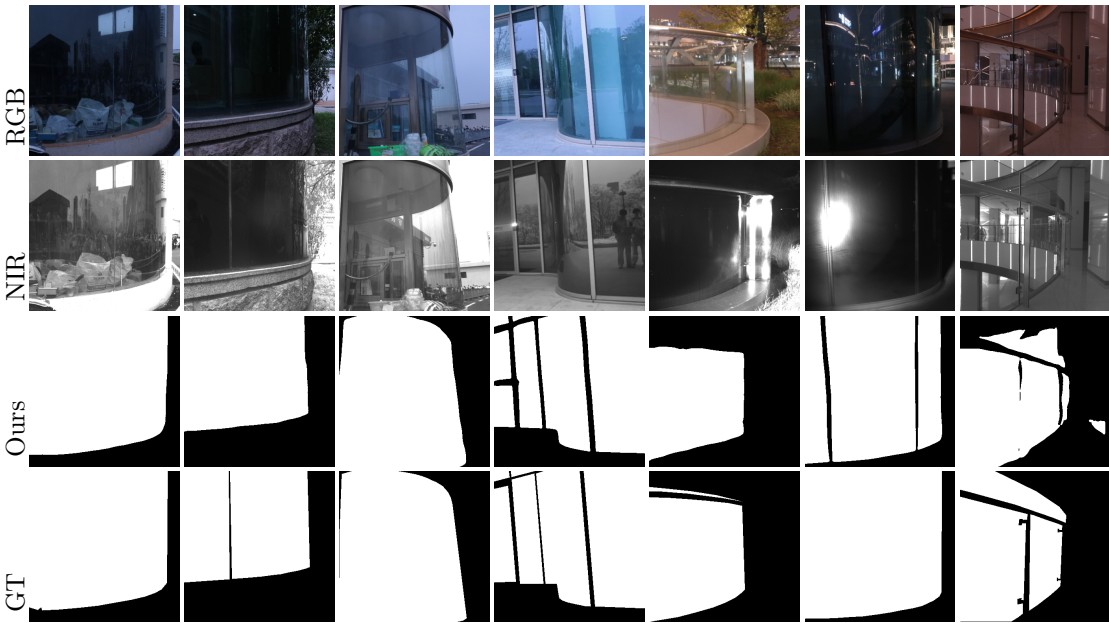

Figure 25: Qualitative results of our method on curved glass surfaces.

### A.10 Study on Dynamic Scenes

Though our method is learned and tested on single images of static scenes, the robustness in dynamic scenes is an important aspect for practical deployment. Below, we discuss the results of my method in dynamic scenes.

1. Sampling frequency discrepancy between RGB and NIR sensors: In our hybrid camera system, the NIR sensor operates at 4.5 FPS while the RGB sensor works at 24 FPS. This inherent frame rate mismatch leads to temporal misalignment when capturing dynamic scenes, which is indeed a challenging setting for fusion-based methods. To evaluate the robustness of our method under real dynamic scenes, we conducted experiments on sequences captured with the moving RGB-NIR camera (as shown in Fig. 26). Despite the severe misalignment caused by different sampling rates, our method can still produce accurate detections. The predicted mask boundaries exhibit only slight distortion, thanks to the rich information provided by the RGB images, which compensates for the temporal misalignment. This demonstrates strong robustness to practical frame-rate mismatches.

2. Motion blur from moving pedestrians/vehicles: We also tested the model on sequences with motion blur (e.g., moving pedestrians). As shown in Fig. 27, motion blur leads to some loss of fine-grained edge details in the predicted masks. However, the overall detection remains reliable: the glass regions are still correctly localized, and the global performance does not degrade significantly. This suggests that our method is robust to typical levels of motion blur encountered in real-world surveillance and autonomous driving scenarios.

3. Computational latency and real-time synchronization: The NIR sensor's low frame rate (4.5 FPS) imposes a relaxed real-time requirement: the system only needs to process one RGB-NIR pair every 222 ms. Our model's inference time on a single NVIDIA RTX 4090 GPU is well below this threshold (e.g., 109.2 ms per pair). Therefore, computational latency does not become a bottleneck for synchronization. For applications that require higher frame rates (e.g., using a faster NIR sensor), our model can be further optimized via pruning or quantization, which we note as future work.

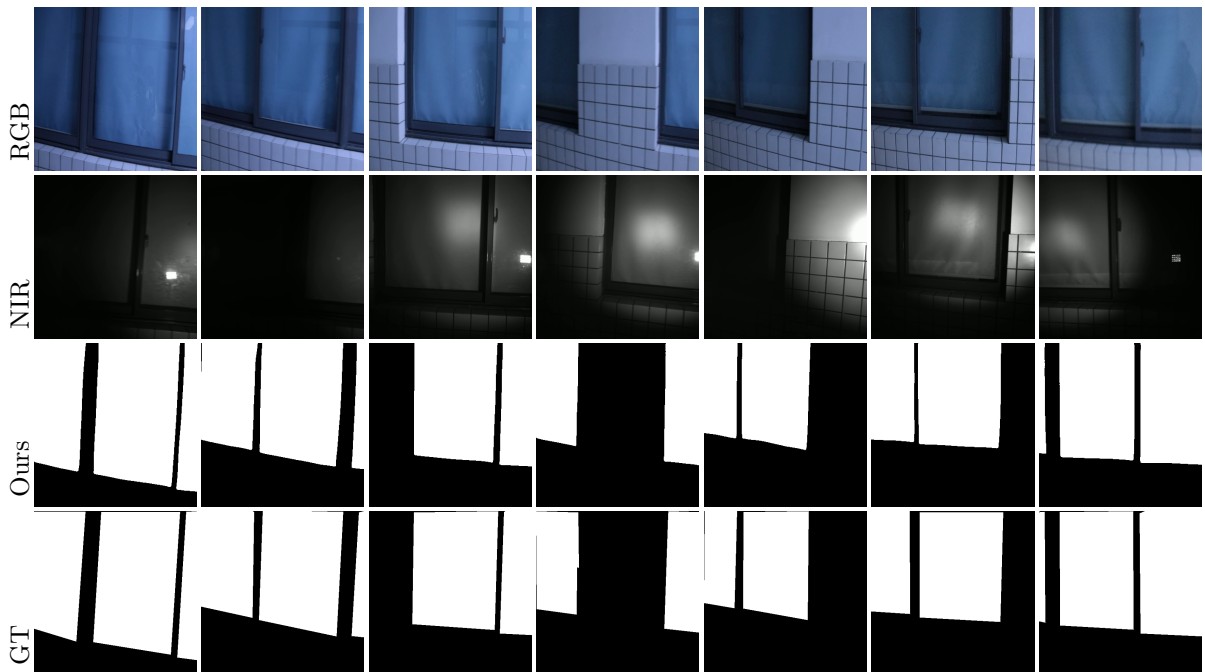

Figure 26: Qualitative results of our model on image sequences captured by a moving camera. To better reflect real-world conditions, all images are not strictly aligned. The ground truth is obtained by labeling in the RGB image.

## A.11    Ablation Study of Retinex Decomposition

As shown in Fig. 28, apart from the case when Retinex Decomposition is not used (*i.e.*, decomposed components are replaced by the original input), we also test the cases when either one of the decomposed components is replaced by the original input image. We show more cases that are not shown in our paper. The reflection on glass surfaces is enhanced in the decomposed Reflectance component ($R_r$) of the RGB image, while there is no easily noticeable reflection on $R_n$. These cases demonstrate that Retinex Decomposition is helpful in distinguishing glass surfaces from non-glass regions.

## A.12    Ablation Study of the Proposed Modules

The GSD task requires better global perception to capture glass surface properties. Previous methods always design modules by integrating features from multiple encoder layers or by applying convolutions with different kernel sizes. Our RNFL module adopts a multi-stage cross-attention design to capture long-range

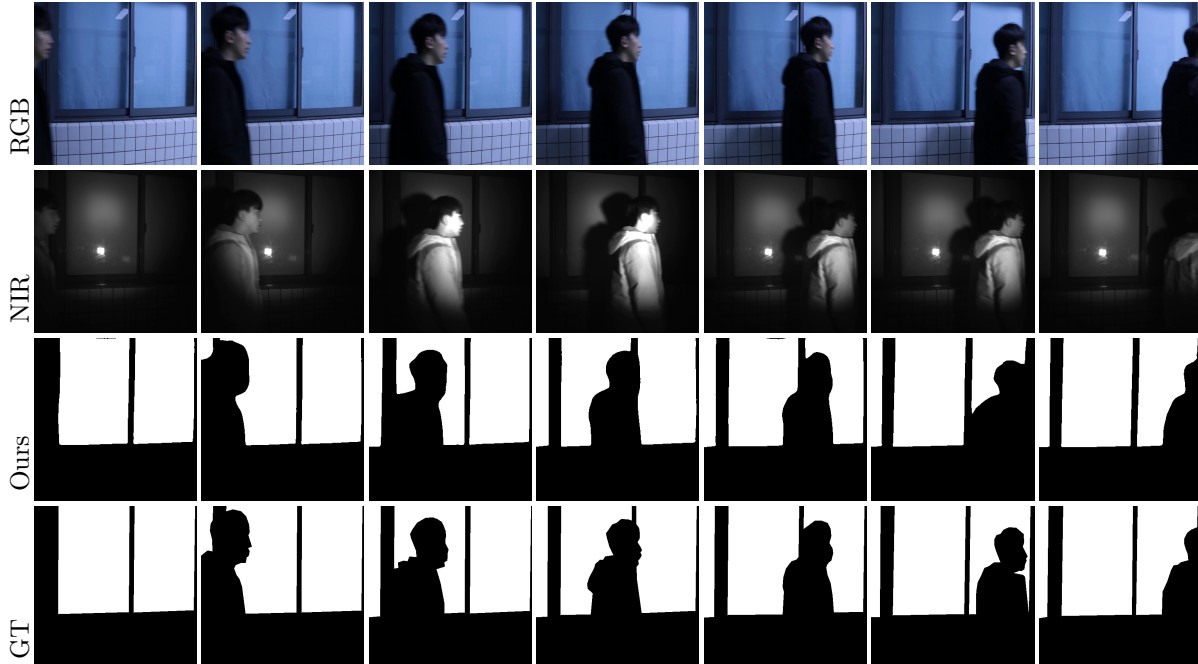

Figure 27: Qualitative results of our model in a scene with a stationary camera and pedestrian movement. To better reflect real-world conditions, all images are not strictly aligned. The ground truth is obtained by labeling in the RGB image.

dependencies and modality-specific cues between RGB/NIR features. In contrast, simple fusion methods, e.g., convolutions, can only provide a limited receptive field, i.e., not very effective in exploiting such relationships.

We ablated RNFL by using a multi-stage convolutional fusion. Specifically, we first concatenate the RGB/NIR features $(X_r^i / \bar{X}_n^i)$ and fuse them using a $3 \times 3$ convolution. The fused features are then concatenated with the decoder features $X_{de}^{i+1}$, followed by another $3 \times 3$ convolution for channel reduction, and finally combined through a residual connection.

This variant leads to an increased model size (50M parameters) and noticeable performance degradation: IoU decreased by 1.28 (87.98→86.70), $F_\beta$ by 0.021 (0.934→0.913), ACC by 0.008 (0.936→0.928), while MAE and BER increased by 0.005 (0.047→0.052) and 0.006 (0.055→0.061), respectively. These results highlight that our attention-based RNFL design is both more accurate and more parameter-efficient.

### A.13 Ablation Study of the ResNet backbone

When the RNGE module is ablated from the variant of our network taking ResNet-50 as the backbone, the performance drop is more pronounced compared to our complete network taking Swin Transformer V2 as the backbone. As shown in Tab. 10, IoU decreased from 82.63 to 78.65, $F_\beta$ dropped from 0.901 to 0.879, ACC declined from 0.905 to 0.883, while MAE increased from 0.070 to 0.086 and BER increased from 0.081 to 0.099.

Table 10: Ablation study of the RNGE module on the variant of our network taking ResNet-50 as the backbone.

| Methods | IoU↑ | $F_\beta$↑ | MAE↓ | BER↓ | ACC↑ |
|---|---|---|---|---|---|
| w/o RNGE | 78.65 | 0.879 | 0.086 | 0.099 | 0.883 |
| full (taking ResNet-50 as the backbone) | 82.63 | 0.901 | 0.070 | 0.081 | 0.905 |

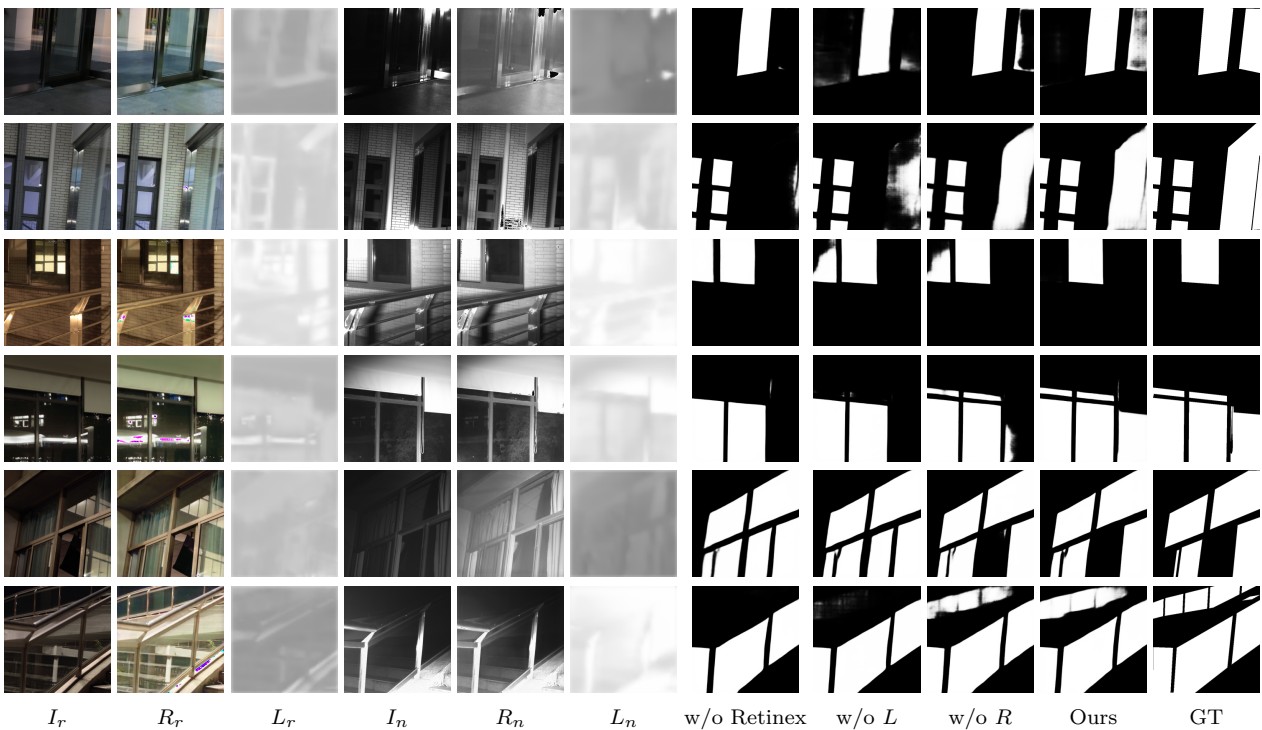

$I_r$    $R_r$    $L_r$    $I_n$    $R_n$    $L_n$    w/o Retinex    w/o $L$    w/o $R$    Ours    GT

Figure 28: Ablation Study on Retinex Decomposition.

