# OpenReview forum: "When Glass Disappears at Night: A Novel NIR-RGB Multimodal Solution"
_TMLR — Accepted by TMLR_

### Review · Reviewer_96gJ · 2026-03-06

**Summary Of Contributions:**

Novel Dataset: The authors constructed the first large-scale nighttime GSD dataset, consisting of 6,192 RGB-NIR image pairs captured in diverse real-world scenes with meticulous manual annotations.

Dual-Branch Network: A novel architecture that utilizes Retinex decomposition to separate reflectance and illumination components from both RGB and NIR modalities.

RNGE Module: The "RGB-NIR Guidance Enhancement" module uses RGB semantic features to refine and enrich NIR reflectance features, specifically aiding in boundary prediction.

RNFL Module: The "RGB-NIR Fusion and Localization" module fuses multi-modal features using gating matrices derived from an Illumination Difference Estimation (IDE) module.

**Audience:**

Yes

**Audience Explanation:**

Quantitative Comparison: The paper provides a comprehensive evaluation against 17 SOTA methods across five metrics (IoU, $F_{\beta}$, ACC, MAE, BER). The proposed method achieves the best performance (e.g., an IoU of 87.98 compared to the next best at 85.66).

Ablation Studies: The effectiveness of specific modules (RNGE and RNFL) is validated. For instance, when RNGE is ablated from a ResNet-50 backbone, the IoU drops from 82.63 to 78.65, proving the module's contribution.

Visual Evidence: The provided figures clearly demonstrate that the proposed method successfully detects glass boundaries that other models either miss or misidentify in low-light conditions.

**Broader Impact Concerns:**

The paper focuses on improving safety and scene understanding for autonomous systems. There are no significant ethical concerns.

**Claims And Evidence:**

Yes

**Claims Explanation:**

Multi-modal Learning: The method of fusing different spectral domains (Visible vs. NIR) via Retinex decomposition and guidance modules provides insights for other cross-modal tasks.

Computer Vision Applications: The research directly benefits the development of robust surveillance systems that operate 24/7.

**Requested Changes:**

The paper lacks discussion on dynamic scene robustness. Specifically, potential issues such as motion blur from moving pedestrians/vehicles, sampling frequency discrepancies between the hybrid sensors, and the impact of computational latency on real-time synchronization are not addressed.

The core premise of the paper relies on the spectral discrepancy between NIR and RGB light. However, the physical properties of glass vary significantly across different types, which is not sufficiently addressed: Common glass variants such as tinted/coated glass (which blocks IR), frosted glass (which scatters light), and stained glass possess vastly different reflectance and transmittance profiles in the NIR spectrum.

---

> ### Author Response · Authors · 2026-04-22
> **Reply to Reviewer(96gJ)**
>
> We thank the reviewer for the careful reading of our manuscript and the constructive suggestions. Our responses to the key concerns are summarized as follows:
>
> **Reviewer(96gJ)Q1**:The paper lacks discussion on dynamic scene robustness. Specifically, potential issues such as motion blur from moving pedestrians/vehicles, sampling frequency discrepancies between the hybrid sensors, and the impact of computational latency on real-time synchronization are not addressed.
>
> **Response**: We thank the reviewer for raising such practical concerns regarding dynamic scenes. Though our method is learned and tested on single images of static scenes, we agree that robustness in dynamic scenes is an important aspect for practical deployment. Below, we address each point based on our new experimental analysis and have integrated the experimental results into Appendix Section A.10 (revised manuscript).
>
> 1. Sampling frequency discrepancy between RGB and NIR sensors:
> In our hybrid camera system, the NIR sensor operates at $4.5$ FPS while the RGB sensor works at $24$ FPS. This inherent frame rate mismatch leads to temporal misalignment when capturing dynamic scenes, which is indeed a challenging setting for fusion‑based methods.
> To evaluate robustness of our method under real dynamic scenes, we have conducted experiments on sequences captured with the moving RGB-NIR camera (as shown in Fig. 26 of the revised manuscript). Despite the severe misalignment caused by different sampling rates, our method can still produce accurate detections. This suggests that the proposed method is robust to practical frame-rate mismatches.
>
> 2. Motion blur from moving pedestrians/vehicles:
> We have also tested the model on sequences with motion blur (e.g., moving pedestrians). As shown in Fig. 27 of the revised manuscript, motion blur leads to some loss of fine-grained edge details in the predicted masks. However, the overall detection remains reliable: the glass regions are still correctly localized, and the global performance does not degrade significantly. This suggests that our method is robust to typical levels of motion blur encountered in real-world surveillance and autonomous driving scenarios.
>
> 3. Computational latency and real‑time synchronization:
> The NIR camera’s low frame rate ($4.5$ FPS) imposes a relaxed real‑time requirement: the system only needs to process one RGB‑NIR pair every ~222 ms. Our model’s inference time on a single NVIDIA RTX 4090 GPU is well below this threshold (e.g., ~109.2 ms per pair). Therefore, computational latency does not become a bottleneck for synchronization. For applications that require higher frame rates (e.g., using a faster NIR sensor), our model can be further optimized via pruning or quantization.
>
> **Reviewer(96gJ)Q2**:The core premise of the paper relies on the spectral discrepancy between NIR and RGB light. However, the physical properties of glass vary significantly across different types, which is not sufficiently addressed: Common glass variants such as tinted/coated glass (which blocks IR), frosted glass (which scatters light), and stained glass possess vastly different reflectance and transmittance profiles in the NIR spectrum.
>
> **Response**: As suggested, we have conducted experiments on multiple glass variants, including tinted glass (which partially blocks IR), frosted glass (which scatters light), and coated glass (using low‑E coatings). The newly added qualitative results are exhibited in Fig. 24 of our supplemental. These results demonstrate that our model maintains accurate glass detection across all tested glass types, despite their different physical properties and NIR responses. Our model does not rely on a fixed NIR reflectance threshold; instead, it learns the relative spectral discrepancy between RGB and NIR modalities.
> The IDE module dynamically adjusts the fusion weights between RGB and NIR modalities. This allows the system to adaptively prioritize the more informative modality, thereby mitigating sensitivity to specific hardware or material-induced intensity variations.
> We have added this discussion to Appendix Section A.8 (revised manuscript).

---

### Review · Reviewer_ywps · 2026-03-16

**Summary Of Contributions:**

[Summary]

This paper addresses the significant challenge of nighttime Glass Surface Detection (GSD) by proposing a multi-modal solution that leverages the complementary visual patterns captured by RGB and Near-Infrared (NIR) sensors. The authors introduce the first large-scale nighttime GSD dataset, consisting of 6,192 carefully annotated RGB-NIR image pairs across 12 diverse real-world scene types. They propose a novel neural network architecture featuring a Retinex-based decomposition stage and two specialized modules: the RGB-NIR Guidance Enhancement (RNGE) module for feature enrichment and the RGB-NIR Fusion and Localization (RNFL) module for controlled feature integration. Extensive experiments demonstrate that the proposed method significantly outperforms 17 state-of-the-art methods in nighttime detection while maintaining robust generalization to daytime scenes.

[Strengths]

* The proposed method aims at addressing nighttime glass detection, where standard RGB, depth, and thermal-based methods frequently fail.
* The provided dataset of over 6,000 carefully annotated triplets is a significant contribution that establishes a high-quality benchmark for future multi-modal vision research.
* Adopting Retinex decomposition to separate reflectance from illumination allows for a more principled approach to modeling material properties under varying light sources.

[Weaknesses]

* The authors acknowledge that the system can suffer from over-detection in extreme low-light scenarios, where it sometimes misidentifies open windows as glass due to insufficient visual cues in the grayscale NIR domain.
* Limited quantitative analysis regarding how the precision of the binocular alignment impacts the performance of the fine-grained cross-attention fusion modules.
* Missing important implementation details and comparison experiments.

**Audience:**

Yes

**Audience Explanation:**

The specific techniques for fusing reflectance and illumination features through cross-attention and gating matrices offer valuable insights for general multi-modal learning. Besides, detecting transparent obstacles is a safety-critical task for mobile robots operating in low-light environments, making this solution directly applicable to navigation problems.

**Claims And Evidence:**

No

**Claims Explanation:**

* Lack of Evidence for the Retinex Decomposition (CTDN) Effectiveness.
A core claim of the paper is that decomposing images into reflectance and illumination components helps the network handle nighttime lighting discrepancies. While the paper provides a quantitative ablation in Table 2, it fails to provide visual samples of the decomposed layers (Reflectance and Illumination maps) generated by the CTDN stage. In nighttime scenarios, where signal-to-noise ratios are low and active NIR creates harsh highlights, Retinex decomposition is notoriously unstable. Without seeing the output of this stage, it is impossible to verify if the network is learning meaningful physical properties or if the decomposition stage is merely acting as a non-linear pre-processing filter.
* Ambiguity in SOTA Comparisons and Retraining.
The paper compares the proposed method against several SOTA methods. However, the submission lacks clear documentation on whether these competing methods (originally designed for daytime RGB or RGB-D) were retrained from scratch on the new nighttime dataset using the same training/test splits. If the SOTAs were not retrained and optimized for the NIR-RGB modality, the "significant improvement" claimed by the authors is technically trivial. If they were retrained, the authors must provide the specific hyperparameter settings to ensure a fair comparison.

**Requested Changes:**

Misalignment Sensitivity: Please include an analysis or discussion on how spatial misalignment between the RGB and NIR sensors affects detection performance, as the fusion modules rely on spatial correspondence.

Failure Case Integration: The failure cases related to over-detection in open windows (currently in the appendix) should be integrated into the main discussion to provide a more complete assessment of the model's limitations.

Active NIR Discussion: Briefly discuss if the method is sensitive to the specific wavelength of the NIR light (850nm vs 940nm) to determine if it is robust to different hardware specifications.

Missing Implementation Details and Comparison Experiments. Please refer to the section above for more details.

---

> ### Author Response · Authors · 2026-04-22
> **Reply to Reviewer(ywps)(1/2)**
>
> We thank the reviewer for the valuable comments and constructive suggestions. Our concise responses are as follows:
>
> **Reviewer(ywps)Q1**: Misalignment Sensitivity: Please include an analysis or discussion on how spatial misalignment between the RGB and NIR sensors affects detection performance, as the fusion modules rely on spatial correspondence.
>
> **Response**: We thank this reviewer for raising this important issue. We agree that spatial misalignment between RGB and NIR sensors is a critical factor for fusion-based methods, as the fusion modules rely on spatial correspondence.
>
> In our method, the RNGE module adopts element-wise subtraction of feature maps to capture complementary cues, while the RNFL module performs element-wise addition for fusion. Both operations inherently assume well-aligned features to ensure stable and accurate fusion. To quantitatively assess the impact of spatial (pixel-level) misalignment, we have conducted experiments introducing artificial pixel offsets between the RGB and NIR inputs. As detailed in Section A.6 (Table 8, in our revised manuscript), the results show that larger offsets (about 10 pixels for the image resolution $384 \times 384$) lead to more performance degradation (IoU drops by 2.4), yet the model maintains reasonable robustness. Notably, spatial misalignment primarily introduces boundary perturbations in the predicted masks, as the model outputs a single fused mask without distinguishing which modality contributes to the detection. To further illustrate this effect, we have provided additional qualitative results in Figs. 23, 26, and 27 of the revised manuscript.
>
> **Reviewer(ywps)Q2**: Failure Case Integration: The failure cases related to over-detection in open windows (currently in the appendix) should be integrated into the main discussion to provide a more complete assessment of the model's limitations.
>
> **Response**: As suggested, we have moved the failure case analysis from the appendix to the main text (Section 5.4 "Limitations" of the revised manuscript) and conducted a more detailed analysis. Some selected failure cases are also shown at the end of Section 5 "Results and Discussion".
>
> **Reviewer(ywps)Q3**: Active NIR Discussion: Briefly discuss if the method is sensitive to the specific wavelength of the NIR light (850nm vs 940nm) to determine if it is robust to different hardware specifications.
>
> **Response**: We agree that a systematic evaluation across different NIR wavelengths would further strengthen the claim of hardware robustness. However, the NIR band setting is fixed internally by the camera hardware and cannot be adjusted by the user.
> In our imaging system, the wavelength of the NIR light (emitted by the active infrared light source) is aligned with the camera’s fixed spectral sensitivity, operating in the $850\sim940$nm band.
> To clarify this point, we have revised our manuscript to explicitly state that the NIR wavelength is constrained by the hardware configuration rather than manually selected.
> Nevertheless, our method is not expected to be sensitive to a precise wavelength within this range.
>
> We have evaluated the generalization ability of our method on the existing daytime RGB-NIR GSD dataset. Results shown in Fig.15 (former Fig.14) and Tab.3 in our revision have demonstrated that our method shows reasonable transfer on daytime scenes. While the wavelength of NIR light typically ranges from $760$nm to $3000$nm, in this daytime RGB-NIR GSD dataset, an NIR filter is used for capturing (placed over a DSLR camera lens, which is different from our setup), which makes the camera operate in the $800 \sim 1100$nm band.

---

> ### Author Response · Authors · 2026-04-22
> **Reply to Reviewer(ywps)(2/2)**
>
> **Reviewer(ywps)Q4**: Lack of Evidence for the Retinex Decomposition (CTDN) Effectiveness. A core claim of the paper is that decomposing images into reflectance and illumination components helps the network handle nighttime lighting discrepancies. While the paper provides a quantitative ablation in Table 2, it fails to provide visual samples of the decomposed layers (Reflectance and Illumination maps) generated by the CTDN stage. In nighttime scenarios, where signal-to-noise ratios are low and active NIR creates harsh highlights, Retinex decomposition is notoriously unstable. Without seeing the output of this stage, it is impossible to verify if the network is learning meaningful physical properties or if the decomposition stage is merely acting as a non-linear pre-processing filter.
>
> **Response**: As suggested, we provide visual samples of the decomposed (reflectance and illumination) layers in Fig. 17 and Fig. 28 (former Fig. 21) of our supplemental. These results demonstrate CTDN's effectiveness and good generalization.
>
> The main reasons are (1) CTDN performs Retinex decomposition within the latent space instead of the image space, which is less sensitive to noise; (2) CTDN is trained on the SICE dataset that contains multi-exposure images, which makes it robust to diverse illumination; and (3) in our method, we directly use the pre-trained CTDN module for RGB decomposition, and finetune the CTDN module for NIR decomposition using the NIR images of our dataset, as claimed in Section 4.4 of the revision.
>
> **Reviewer(ywps)Q5**: Ambiguity in SOTA Comparisons and Retraining. The paper compares the proposed method against several SOTA methods. However, the submission lacks clear documentation on whether these competing methods (originally designed for daytime RGB or RGB-D) were retrained from scratch on the new nighttime dataset using the same training/test splits. If the SOTAs were not retrained and optimized for the NIR-RGB modality, the ``significant improvement" claimed by the authors is technically trivial. If they were retrained, the authors must provide the specific hyperparameter settings to ensure a fair comparison.
>
> **Response**: Thanks for the comment. All competing methods were retrained on our dataset using the same training/test splits, and following their original settings of hyperparameters (e.g., optimizer, learning-rate schedule, epochs). We have added these details to section 5.1 of the revised manuscript.

---

> ### Comment · Action_Editor_VS9r · 2026-05-24
> **Please submit your final Decision Recommendation as soon as possible.**
>
> Dear #Reviewer ywps,
>
> The authors have provided a rebuttal for your comments. Please read the rebuttal and submit your final decision as soon as possible.
>
> Regards from the Action Editor.

---

### Review · Reviewer_ATEp · 2026-04-12

**Summary Of Contributions:**

This paper studies nighttime glass surface detection, a setting that has received much less attention than daytime GSD. The main idea is that RGB and active NIR images provide complementary cues at night, since glass can exhibit different reflections, edges, and illumination patterns across the two modalities. Building on this observation, the paper makes three main contributions: it introduces a new nighttime RGB–NIR glass detection dataset with 6,192 aligned image pairs and mask annotations; it proposes a decomposition-based dual-modal network with two dedicated modules, RNGE and RNFL, to enhance NIR features with RGB guidance and to fuse the modalities using illumination-aware gating; and it provides an extensive empirical study showing strong performance on the proposed benchmark, along with ablations and transfer experiments to daytime settings.

The problem is practically meaningful, especially for surveillance and robotics settings that operate at night. The dataset appears to be a genuine contribution, and the experiments are broad: the authors compare against many retrained baselines, include detailed ablations, study alternative modalities, and discuss failure cases. The main technical idea is also physically motivated rather than purely heuristic.

The main weaknesses are more about positioning and presentation than about the central contribution. First, the claim that the method “generalizes well” to daytime scenes feels a bit too strong, since on the daytime RGB–NIR benchmark it underperforms NRGlassNet on all reported metrics. Second, although the ablations do support the architecture, the gains from individual modules are relatively modest, so a bit more discussion would help readers understand where the improvements are coming from. Third, the dataset seems mainly focused on smooth, flat glass, which may limit generality. Finally, some comparisons in the appendix, especially Table 9, are incomplete.

**Audience:**

Yes

**Audience Explanation:**

I believe this paper would be of interest to at least part of the TMLR audience. Nighttime perception is an important practical problem, and glass detection is relevant to robotics, scene understanding, and safety-related applications. The paper also introduces what appears to be the first dedicated nighttime RGB–NIR GSD dataset, which by itself may be useful to researchers working on multimodal perception or glass/transparent object understanding.

**Claims And Evidence:**

Yes

**Claims Explanation:**

I think the main claims are supported by the evidence in the paper. The central claim is that the proposed RGB–NIR method performs better than existing alternatives for nighttime glass surface detection, and Table 1 supports this clearly: the method achieves the best results among the compared baselines on all five reported evaluation metrics on the proposed dataset. The qualitative examples are also consistent with the quantitative results and show improvements in difficult nighttime scenes.

My only reservation is that one part of the framing is a bit stronger than the evidence really supports. The paper says the method generalizes well to daytime scenes, but on the daytime RGB–NIR benchmark in Table 3 it trails NRGlassNet on every reported metric. I do think the results show that the method remains reasonably competitive outside the nighttime setting, but I would encourage the authors to phrase that claim more carefully. This does not undermine the main experimental evidence for the nighttime contribution, which I found solid overall.

**Requested Changes:**

1. Adjust the claim about daytime generalization. On the daytime RGB–NIR benchmark in Table 3, the proposed method performs worse than NRGlassNet on all reported metrics. I would suggest replacing phrases such as “generalizes well to daytime scenes” with something more measured, such as “shows reasonable transfer” or “remains competitive,” and briefly discussing this limitation in the main text.

2. Complete or revise Table 9 in the appendix. The comparison to TransLab contains several missing entries, which makes the comparison harder to interpret. If these metrics are unavailable, it would be better to either explain this explicitly in the caption/text or restrict the table to shared metrics only.

3. Discuss dataset bias and scope more explicitly in the main paper. The paper notes that glass regions tend to cluster in the upper part of the image and that most examples involve smooth, flat glass. Both points are important, since they may affect how well the learned model transfers to more varied settings.

4. It would help to add a bit more interpretation around the ablations. The module-wise gains are consistent but fairly small, so a short discussion of how the modules interact, or whether their benefits are complementary rather than additive, would make the design choices easier to appreciate.

5. Please make the implementation details around Retinex fine-tuning a bit easier to follow. The appendix includes useful training details, but it is still somewhat unclear which settings apply specifically to CTDN fine-tuning versus the second-stage detector training. Clarifying this would improve reproducibility.

6. I would encourage the authors to move at least one of the practical limitations from the appendix into the main paper, especially the fact that the dataset mostly contains smooth/flat glass and the failure cases under very weak RGB cues or poor decomposition.

---

> ### Author Response · Authors · 2026-04-22
> **Reply to  Reviewer(ATEp) (1/2)**
>
> We would like to thank you for the insightful comments for our work. Below, we address the concerns raised in the feedback.
>
> **Reviewer(ATEp)Q1**:  Adjust the claim about daytime generalization. On the daytime RGB–NIR benchmark in Table 3, the proposed method performs worse than NRGlassNet on all reported metrics. I would suggest replacing phrases such as “generalizes well to daytime scenes” with something more measured, such as “shows reasonable transfer” or “remains competitive,” and briefly discussing this limitation in the main text.
>
> **Response**: Thanks for the suggestion. We have modified "...generalizes well to daytime scenes'' to "...maintaining competitive performance in daytime conditions".
>
> **Reviewer(ATEp)Q2**:  Complete or revise Table 9 in the appendix. The comparison to TransLab contains several missing entries, which makes the comparison harder to interpret. If these metrics are unavailable, it would be better to either explain this explicitly in the caption/text or restrict the table to shared metrics only.
>
> **Response**: As suggested, we restrict Tab.9 (in our revision) to shared metrics for clarity, as the missing entries are not available due to different evaluation settings.
>
> **Reviewer(ATEp)Q3**:  Discuss dataset bias and scope more explicitly in the main paper. The paper notes that glass regions tend to cluster in the upper part of the image and that most examples involve smooth, flat glass. Both points are important, since they may affect how well the learned model transfers to more varied settings.
>
> **Response**: As suggested, we discuss dataset bias and scope explicitly (Section 3 "Proposed Dataset" of the revised manuscript).
> In our dataset, glass surfaces are predominantly smooth and flat, and their spatial distribution tends to concentrate in the upper regions of the images. This reflects common real-world configurations (e.g., windows and doors installed around or above eye level), but may introduce bias. In particular, it could limit generalization to more diverse scenarios, such as curved glass, or cases where glass appears in atypical spatial locations.
>
> Nonetheless, our method maintains stable performance across different scenes in our evaluation, as illustrated in Fig. 25 of the revised manuscript.
> Expanding the dataset to include more diverse glass types (e.g., curved, fractured, or heavily textured surfaces) and spatial distributions is an important direction for future work.
>
> **Reviewer(ATEp)Q4**:  It would help to add a bit more interpretation around the ablations. The module-wise gains are consistent but fairly small, so a short discussion of how the modules interact, or whether their benefits are complementary rather than additive, would make the design choices easier to appreciate.
>
> **Response**: We thank the reviewer for this insightful suggestion. We would like to emphasize that the proposed modules are designed to be complementary rather than purely additive, with each addressing a different aspect of the problem:
>
> RNGE focuses on enhancing informative reflectance cues by leveraging cross-modal differences, particularly improving NIR representations under weak RGB conditions.
>
> IDE provides illumination-aware guidance derived from Retinex decomposition, which helps regulate the relative contribution of RGB and NIR features.
>
> RNFL performs progressive multi-scale fusion, integrating the enhanced reflectance features under the guidance of illumination differences.
>
> We have included a brief discussion around the ablations (Subsection 5.3  ``Ablation Results").
> While the performance gains from individual modules appear to be relatively modest, they are consistent across settings, indicating stable and complementary contributions.
> In addition, we further test the effectiveness of our RNGE module against a less powerful backbone (ResNet-50), where we observe a significant performance drop by ablating the RNGE module. As shown in the following table, IoU decreased from $82.63$ to $78.65$, $F_\beta$ dropped from $0.901$ to $0.879$, ACC declined from $0.905$ to $0.883$, while MAE increased from $0.070$ to $0.086$ and BER increased from $0.081$ to $0.099$.
>
> **Table:** Ablation study of the RNGE module on the variant of our network with a ResNet-50 backbone.
>
> | Methods                                  | IoU ↑ | Fβ ↑  | MAE ↓ | BER ↓ | ACC ↑ |
> |------------------------------------------|-------|-------|-------|-------|-------|
> | w/o RNGE                                 | 78.65 | 0.879 | 0.086 | 0.099 | 0.883 |
> | Full model (ResNet-50 backbone)          | 82.63 | 0.901 | 0.070 | 0.081 | 0.905 |

---

> ### Author Response · Authors · 2026-04-22
> **Reply to Reviewer(ATEp) (2/2)**
>
> **Reviewer(ATEp)Q5**: Please make the implementation details around Retinex fine-tuning a bit easier to follow. The appendix includes useful training details, but it is still somewhat unclear which settings apply specifically to CTDN fine-tuning versus the second-stage detector training. Clarifying this would improve reproducibility.
>
> **Response**: As suggested, we have revised our manuscript to clarify this point.
> Specifically, our training pipeline consists of two separate stages, which follow a consistent training protocol as described in Sec. A.1:
>
> Stage 1 (Retinex decomposition / CTDN fine-tuning):
> We initialize the CTDN module with pre-trained weights and fine-tune it for NIR decomposition using the reconstruction loss $L_{rec}$. During this stage, only the CTDN parameters are updated, while the rest of the network remains inactive.
>
>
>
> Stage 2 (Glass detection network training):
> After Stage 1, the CTDN module is frozen, and we train the encoder–decoder network for glass surface and boundary detection using $L_{stage2}$.
>
> We have refined the implementation details in order to clearly separate these two stages and explicitly list their corresponding training settings to improve clarity and reproducibility.
>
> **Reviewer(ATEp)Q6**: I would encourage the authors to move at least one of the practical limitations from the appendix into the main paper, especially the fact that the dataset mostly contains smooth/flat glass and the failure cases under very weak RGB cues or poor decomposition.
>
> **Response**: Thanks for the suggestion. We have added a Subsection 5.5 "Limitations" at the end of Section 5 "Results and Discussion" in the main paper.

---

### Decision · Action_Editor_VS9r · 2026-05-25

**Recommendation:** Accept as is

**Additional Comments:**

Please revise the paper in accordance with the reviewers' comments before submitting the final version.

**Audience:**

Yes

**Audience Explanation:**

The paper makes a solid contribution on both the dataset and methodological fronts.

**Claims And Evidence:**

Yes

**Claims Explanation:**

Most of the claims in this manuscript have been supported by accurate, convincing, and clear evidence.

Besides, all the reviewers indicated that their initial concerns have been well addressed, and gave a Learning Accept recommendation for this paper.